



# Climate controlled root zone parameters show potential to improve water flux simulations by land surface models

Fransje van Oorschot[1,2], Ruud J. van der Ent[1], Markus Hrachowitz[1], and Andrea Alessandri[2,3]

[1]Department of Water Management, Faculty of Civil Engineering and Geosciences, Delft University of Technology, Delft, The Netherlands
[2]Royal Netherlands Meteorological Institute (KNMI), De Bilt, The Netherlands
[3]Institute of Atmospheric Sciences and Climate, National Research Council of Italy (CNR-ISAC), Bologna, Italy

**Correspondence:** Fransje van Oorschot (f.vanoorschot@tudelft.nl)

**Abstract.** The root zone storage capacity ($S_r$) is the maximum volume of water in the subsurface that can potentially be accessed by vegetation for transpiration. It influences the seasonality of transpiration as well as fast and slow runoff processes. Many studies have shown that $S_r$ is heterogeneous as controlled by local climate conditions, which affect vegetation strategies in sizing their root system able to support plant growth and to prevent water shortages. Root zone parameterization in most land surface models does not account for this climate control on root development, being based on look-up tables that prescribe worldwide the same root zone parameters for each vegetation class. These look-up tables are obtained from measurements of rooting structure that are scarce and hardly representative of the ecosystem scale. The objective of this research is to quantify and evaluate the effects of a climate controlled representation of $S_r$ on the water fluxes modeled by the HTESSEL land surface model. Climate controlled $S_r$ is here estimated with the "memory method" (MM) in which $S_r$ is derived from the vegetation's memory of past root zone water storage deficits. $S_{r,MM}$ is estimated for 15 river catchments over Australia across three contrasting climate regions: tropical, temperate and Mediterranean. Suitable representations of $S_{r,MM}$ are implemented in an improved version of HTESSEL (MD) by accordingly modifying the soil depths to obtain a model $S_{r,MD}$ that matches $S_{r,MM}$ in the 15 catchments. In the control version of HTESSEL (CTR), $S_{r,CTR}$ is larger than $S_{r,MM}$ in 14 out of 15 catchments. Furthermore, the variability among the individual catchments of $S_{r,MM}$ (117–722 mm) is considerably larger than of $S_{r,CTR}$ (491–725 mm). The climate controlled representation of $S_r$ in the MD version results in a significant and consistent improvement of the modeled monthly seasonal climatology (1975–2010) and inter-annual anomalies of river discharge compared with observations. However, the effects on biases in long-term annual mean fluxes are small and mixed. The modeled monthly seasonal climatology of the catchment discharge improved in MD compared to CTR: the correlation with observations increased significantly from 0.84 to 0.90 in tropical catchments, from 0.74 to 0.86 in temperate catchments and from 0.86 to 0.96 in Mediterranean catchments. Correspondingly, the correlations of the inter-annual discharge anomalies improve significantly in MD from 0.74 to 0.78 in tropical catchments, from 0.80 to 0.85 in temperate catchments and from 0.71 to 0.79 in Mediterranean catchments. The results indicate that the use of climate controlled $S_{r,MM}$ can significantly improve the timing of modeled discharge and, by extension, also evaporation fluxes in land surface models. On the other hand, the method has not shown to significantly reduce long-term climatological model biases over the catchments considered for this study.





## 1 Introduction

Vegetation controls the partitioning of precipitation into evaporation and runoff by transporting water through their roots to the atmosphere and is thereby key in the representation of land surface-atmosphere interactions (Milly, 1994; Seneviratne et al., 2010). The moisture flow from the land surface to the atmosphere through vegetation root water uptake is defined as transpi-

ration and is globally the largest water flux from terrestrial ecosystems (Schlesinger and Jasechko, 2014). The contribution of transpiration to total land evaporation is regulated by the interplay between the atmospheric water demand and the soil moisture within the reach of vegetation's roots. The root zone is defined as the part of the subsurface where vegetation has developed roots and can be characterized by parameters such as root depth and root density. The importance of the root zone in land surface and climate modelling is widely acknowledged and multiple studies emphasize the climate sensitivity to changes in the

vegetation's root zone (Mahfouf et al., 1996; Desborough, 1997; Zeng et al., 1998; de Rosnay and Polcher, 1998; Norby and Jackson, 2000; Feddes et al., 2001; Teuling et al., 2006). However, the parameterization of the root zone in state-of-the-art land surface models (LSMs) is a possible cause for the large uncertainties in water flux representations in these models (Gharari et al., 2019), which is in particular true for land evaporation simulations (Pitman, 2003; Seneviratne et al., 2006; Wartenburger et al., 2018).

The hydrologically relevant magnitude of the vegetation's root zone can be described by the root zone water storage capacity $S_r$, that represents the maximum subsurface moisture volume that can be accessed by the vegetation's roots. The size of $S_r$ controls the variability and timing of water fluxes and specifically the ability of vegetation to maintain transpiration during the dry season when there is little to no recharge (Milly, 1994). It is important to note that $S_r$ is not necessarily proportional to the depth of roots. While root depth only describes the vertical root profile, $S_r$ also accounts for lateral root extent as well as root

density. For example, an ecosystem covered by deep rooting vegetation with roots with low density likely has a smaller $S_r$ than one covered by vegetation with shallow, high density roots (Singh et al., 2020).

However, most global LSMs do not have the explicit objective to estimate $S_r$ and rather aim for a description of root zone parameters (e.g. root depth, root density and root distribution) for different vegetation classes combined with soil type information and a model-dependent fixed soil depth. The generally shallow ($< 2\,\mathrm{m}$) (Pan et al., 2020) fixed soil depth limits

the size of $S_r$ and, as a consequence, also the moisture extraction by roots from deep soil layers (Kleidon and Heimann, 1998; Sakschewski et al., 2020). LSMs use look-up tables that prescribe worldwide the same root zone parameters for each combination of vegetation and soil class as obtained from a very limited number of point-scale observations of rooting structure (Canadell et al., 1996; Jackson et al., 1996; Zeng et al., 1998; Schenk and Jackson, 2002a, b). The spatial distribution of the root zone parameterization in LSMs is obtained by combining these look-up table values with maps of vegetation cover and

soil texture. The limitations of this approach are as follows: the root observations are 1) uncertain due to the fact that they mostly vertically extrapolate root measurements while excavating only the first meter or less (Schenk and Jackson, 2002a, b), 2) do not adequately represent global distributions of root structures because observations are extremely scarce: e.g. the Schenk





and Jackson (2002b) dataset includes 475 root profiles in 209 geographical locations, 3) observations of individual plants that do not represent spatial variations in ecosystem composition at scales larger than the plot scale and 4) snapshots in time and,

therefore, do not represent their evolution over time due to continuous adaptation of ecosystems to changing environmental conditions.

An alternative to the look-up tables based on point-scale root observations for describing the vegetation's root zone is a climate controlled approach. The only LSM to our knowledge in which climate controlled root zone parameters are used is the JSBACH3.2 model (Hagemann and Stacke, 2015) in which rooting depths are based on the optimisation model of net primary

production from Kleidon (2004). Yet, there is general strong evidence that climate is the dominant control of root development in many environments, as vegetation tends to optimize its above- and below-ground carbon investment in order to optimally function by avoiding water shortages and maintaining transpiration and productivity (Collins and Bras, 2007; Guswa, 2008; Sivandran and Bras, 2013). For example, it is likely that trees in a dry climate develop a larger $S_r$ than trees in a wet climate because trees in a dry climate need to invest more in growing roots to sustain their water demand (Gentine et al., 2012; Gao

et al., 2014).

A widely applied climate controlled approach in catchment hydrological studies to describe $S_r$ is the "memory method". In this method $S_r$ is derived from water storage deficit calculations in the root zone at catchment scale, assuming vegetation is able to keep memory of past deficit conditions to size roots in such a way to guarantee continuous access to water (hereinafter $S_{r,MM}$) (Gentine et al., 2012; Gao et al., 2014). Recent studies demonstrated that this method provides plausible catchment-

scale estimates of $S_r$ (e.g., Gao et al., 2014; Nijzink et al., 2016; Wang-Erlandsson et al., 2016; Hrachowitz et al., 2020), that result in improvements in modelling catchment discharge compared to soil derived $S_r$ estimates (De Boer-Euser et al., 2016). However, climate controlled root zone parameters have not yet been widely incorporated in LSMs.

The objective of this study is to quantify and evaluate the effects of a climate controlled representation of $S_r$ on the water fluxes modeled by the HTESSEL land surface model. Specifically we will test the hypothesis that implementing $S_{r,MM}$ in

HTESSEL can improve the modeled magnitude and timing of catchment discharge and evaporation fluxes. By applying the memory method for estimating ecosystem-scale $S_r$ for use in LSMs, the first three limitations of using sparse root observations mentioned above can be overcome, but it should be acknowledged that, although the memory method in principle allows to adaptively update $S_r$, in this work we use a fixed value in time. In this study, $S_{r,MM}$ values representative for the 1973–2010 time period are estimated for 15 Australian catchments across different climate regions (Sect. 2.3 and Appendix A). The $S_{r,MM}$

estimates are then used to constrain the $S_r$ in HTESSEL (Sect. 2.5). Section 3 evaluates the effects on discharge and evaporation in HTESSEL by performing offline simulations with and without the improved representation of $S_r$. Finally in Sect. 4 and 5 the potential for a wider application of climate controlled root zone parameters is discussed.

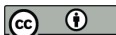



## 2 Methods

### 2.1 Study area

Australia is characterized by large spatial differences in precipitation (Fig. 1), vegetation coverage and temperatures, varying from hot and dry deserts in the interior to tropical forests with a monsoon season in the north. We have selected 15 Australian river catchments with station observations of river discharge at the outlet of the catchment to estimate $S_r$ applying the memory method (Fig. 1; Table S1) (Australian Government Bureau of Meteorology, 2019). The catchments are selected based on available discharge data (at least 30 years of station observations), size (at least one third of the land surface model grid

cell area of approximately $5500\,\mathrm{km}^2$ in order to spatially extrapolate catchment characteristics to grid cells) and differences in climate (spatial spread of the catchments across Australia for the analysis of different climate zones). The catchments are classified in three climate regions based on their hydrological characteristics (Table 1; Fig. 2; Table S2). The tropical catchments are characterized by pronounced seasonality of rainfall with a seasonality index of precipitation ($I_S$) of 0.7 or higher, while temperate and Mediterranean catchments have year-round rainfall ($I_S < 0.7$). The Mediterranean catchments are characterized

by a time-lag $\phi$ between long-term mean maximum monthly potential evaporation $E_p$ and precipitation $P$ of five or six months, while in tropical and temperate catchments mean maximum monthly $E_p$ and $P$ occur within three months.

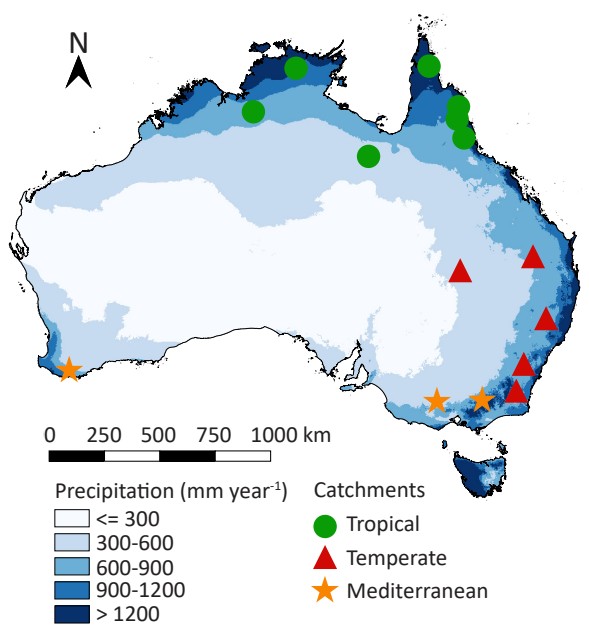

**Figure 1.** Location of the 15 study catchments within Australia. The green, red and orange markers indicate the climate region and the blue shades indicate long-term mean annual precipitation (Australian Government Bureau of Meteorology, 2019). A list of the catchments and their characteristics is provided in Table S1.





**Table 1.** Average hydrological characteristics of the catchments in the three climate regions for the time period 1973–2010 with long-term mean annual discharge $\overline{Q}$, precipitation $\overline{P}$ and potential evaporation $\overline{E_p}$, aridity index $I_A = \overline{E_p}/\overline{P}$, seasonality index of precipitation $I_S = \frac{1}{\overline{P_a}} \sum_{m=1}^{m=12} |\overline{P_m} - \frac{\overline{P_a}}{12}|$, with $\overline{P_a}$ the annual mean precipitation and $\overline{P_m}$ the monthly mean precipitation of month $m$ (Gao et al., 2014) and the time-lag $\phi$ between long-term mean maximum monthly precipitation ($P$) and potential evaporation ($E_p$). Values for all individual catchments are provided in Table S2.

| Climate region | $\overline{Q}$ (mm year$^{-1}$) | $\overline{P}$ (mm year$^{-1}$) | $\overline{E_p}$ (mm year$^{-1}$) | $I_A$ (-) | $I_S$ (-) | $\phi$ (months) |
|---|---|---|---|---|---|---|
| Tropical | 302 | 1101 | 1869 | 2 | 0.9 | 2.3 |
| (7 catchments) | | | | | | |
| Temperate | 57 | 651 | 1488 | 2.5 | 0.2 | 0.6 |
| (5 catchments) | | | | | | |
| Mediterranean | 53 | 879 | 1276 | 1.7 | 0.3 | 5.7 |
| (3 catchments) | | | | | | |

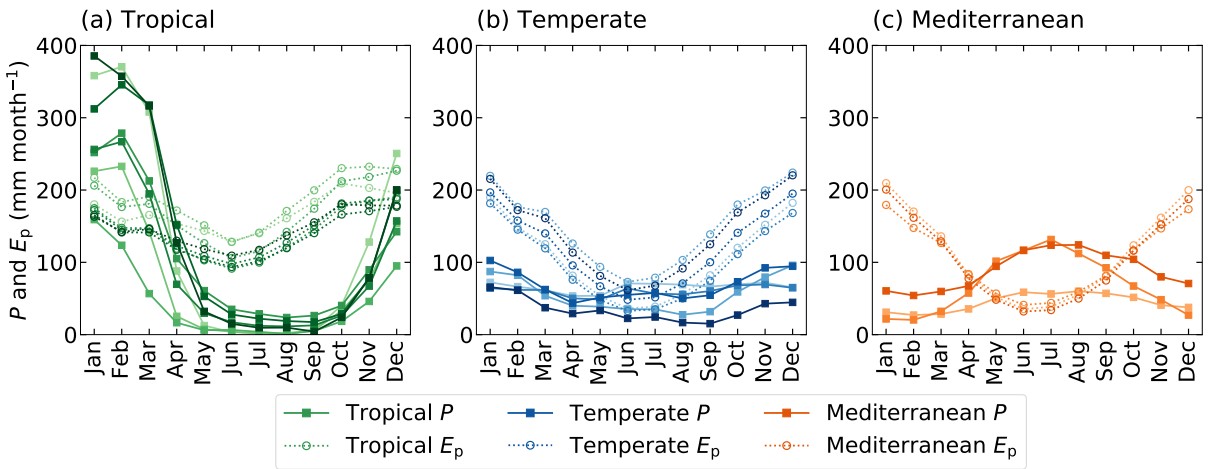

**Figure 2.** Monthly seasonal climatology of precipitation ($P$) and potential evaporation ($E_p$) for the (a) tropical, (b) temperate and (c) Mediterranean catchments with the solid lines $P$ and the dashed lines $E_p$ for the time series 1973–2010. The different shades indicate the 15 individual study catchments.

## 2.2 Data

For this study we use daily discharge data from station observations in the catchments for the time period 1973–2010 (Australian Government Bureau of Meteorology, 2019). For the same time period we use daily precipitation and daily mean temperature data from the GSWP-3 dataset on a regular 0.5°grid (Kim, 2017). Daily $E_p$ is calculated applying the Hargreaves and Samani formulation, based on temperature and radiation (Hargreaves and Samani, 1982; Mines ParisTech Solar radiation Data, 2016). The FLUXCOM RS+METEO dataset is used as a reference dataset to benchmark modeled actual evaporation.





FLUXCOM provides a gridded product of interpolated monthly evaporation as a fusion of FLUXNET eddy covariance towers, satellite observations and meteorological data (GSWP-3) for the time period 1975–2010 (Jung et al., 2019). This dataset has shown plausible estimates of mean annual and seasonal evaporation and is generally considered as a suitable tool for global land model evaluations (Jung et al., 2019; Ma et al., 2020). However, we found considerable differences between the long-term annual mean evaporation $\overline{E}_{\mathrm{FLUXCOM}}$ and $\overline{E}$ derived from the catchment water balance ($\overline{E}_{\mathrm{WB}}$) based on observed $Q$ and GSWP-3 $P$ ($\overline{E}_{\mathrm{WB}} = \overline{P} - \overline{Q}$) (Fig. 3). Figure 3 clearly illustrates that the $\overline{E}_{\mathrm{FLUXCOM}}$ is consistently lower than $\overline{E}_{\mathrm{WB}}$ with an average difference of $150\,\mathrm{mm\,year^{-1}}$, equivalent to about 20 % of the long-term water balances. $\overline{E}_{\mathrm{WB}}$ is likely to be more reliable than $\overline{E}_{\mathrm{FLUXCOM}}$ because $\overline{E}_{\mathrm{WB}}$ provides an integrated catchment scale estimate as it is derived from observations of $Q$. In addition, $\overline{E}_{\mathrm{FLUXCOM}}$ is based on point scale estimates of FLUXNET stations that do not coincide with and are mostly located far from the study catchments (Pastorello et al., 2020). The discrepancy between the FLUXCOM and the catchment water balance is addressed by scaling the monthly FLUXCOM evaporation:

$$E_{\mathrm{FLUXCOM\text{-}WB}} = E_{\mathrm{FLUXCOM}} \frac{\overline{E}_{\mathrm{WB}}}{\overline{E}_{\mathrm{FLUXCOM}}} \tag{1}$$

with $E_{\mathrm{FLUXCOM\text{-}WB}}$ the monthly reference evaporation representative for the catchment scale, $E_{\mathrm{FLUXCOM}}$ from Jung et al. (2019) in the catchment corresponding grid cells and $\frac{\overline{E}_{\mathrm{WB}}}{\overline{E}_{\mathrm{FLUXCOM}}}$ the catchment specific scaling factor.

We use gridded data of vegetation type and coverage derived from the GLCC1.2 (ECMWF, 2016) and soil texture data from the FAO/UNESCO Digital Soil Map of the World (FAO, 2003). Characteristics of the different soil textures are based on the Van Genuchten soil parameters (Van Genuchten, 1980). These data are needed as input of the HTESSEL model and for the estimation of $S_{\mathrm{r}}$.

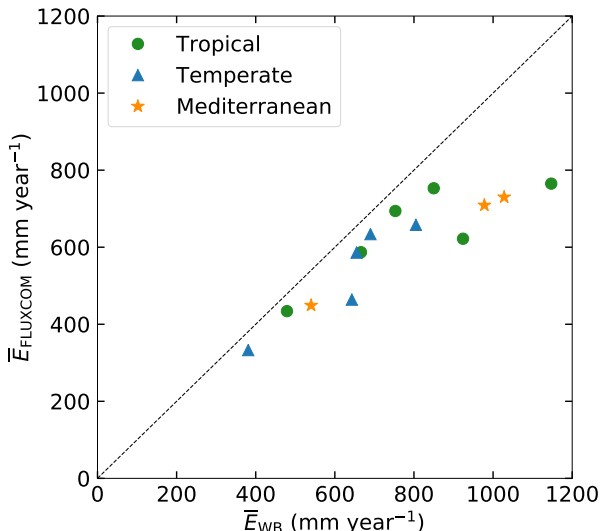

**Figure 3.** Long-term mean annual evaporation ($\overline{\overline{E}}$) as estimated from long-term water balance data ($\overline{E}_{\mathrm{WB}}$) compared to the FLUXCOM dataset ($\overline{E}_{\mathrm{FLUXCOM}}$) for the 1975–2010 period.





### 2.3 Memory method for estimating root zone storage capacity

$S_{r,MM}$ is estimated based on catchment hydrometeorological data, according to the methodology described in the studies of De Boer-Euser et al. (2016), Nijzink et al. (2016) and Wang-Erlandsson et al. (2016). The cumulative water storage deficit ($S_d$ (mm) in the root zone is based on daily time series of effective precipitation $P_e$ ($mm\,day^{-1}$) and transpiration $E_t$ ($mm\,day^{-1}$) for the time period $t_0$–$t_1$ of 1973–2010 and is described by:

$$S_d(t) = \max(0, -\int_{t_0}^{t_1}(P_e - E_t)dt) \tag{2}$$

where $P_e$ ($mm\,day^{-1}$) is derived from the water balance of the interception storage $S_i$:

$$\frac{dS_i}{dt} = P - E_i - P_e \tag{3}$$

with $P$ the precipitation ($mm\,day^{-1}$) and $E_i$ the interception evaporation ($mm\,day^{-1}$). Equation 3 is solved by the following relations:

$$E_i = \begin{cases} E_p & \text{if } E_p dt < S_i \\ \frac{S_i}{dt} & \text{if } E_p dt \geq S_i \end{cases} \tag{4}$$

$$P_e = \begin{cases} 0 & \text{if } S_i \leq S_{i,max} \\ \frac{S_i - S_{i,max}}{dt} & \text{if } S_i > S_{i,max} \end{cases} \tag{5}$$

where $E_p$ is the potential evaporation ($mm\,day^{-1}$) and $S_{i,max}$ the maximum interception storage (mm). $S_{i,max}$ depends on the land cover, and is estimated between $2 - 8\,mm$ for a tropical forest (Herwitz, 1985) and between $0 - 3\,mm$ for a temperate forest (Gerrits et al., 2010). However, De Boer-Euser et al. (2016) found that the sensitivity of $S_r$ to the value of $S_{i,max}$ is small and, therefore, here a value of $2.5\,mm$ is used in all catchments for simplicity.

Daily $E_t$ ($mm\,day^{-1}$) in Eq. 2 was calculated by:

$$E_t(t) = c\,E_p(t) \tag{6}$$

where $c$ $(-)$ is a coefficient that represents the ratio between transpiration and potential evaporation $c = \overline{E_t}/\overline{E_p}$. $\overline{E_t}$ ($mm\,year^{-1}$) is the long-term mean transpiration derived from the water balance ($\overline{E_t} = \overline{P_e} - \overline{Q}$) and $\overline{E_p}$ ($mm\,year^{-1}$) the long-term mean potential evaporation. The subtle interactions between atmospheric water demand and vegetation-available water supply can lead to inter-annual variability in $c$. The above described approach that provided constant estimates of $c$ is therefore extended by an iterative procedure to estimate annually varying values of coefficient $c$ as described in Appendix A.

Catchment $S_{r,MM}$ (mm) is estimated based on the assumption that a catchment's ecosystem designs its rooting system while keeping memory of water stress events with certain return periods. Previous studies provide evidence that these return periods





are likely to be larger for high vegetation (e.g. forest) than for low vegetation (e.g. grass). Based on the results of Gao et al. (2014), De Boer-Euser et al. (2016) and Wang-Erlandsson et al. (2016) drought return periods (RP) for high and low vegetation are set to 40 and 2 years, respectively. The $S_{r,MM}$ corresponding to these drought return periods is calculated applying the

Gumbel extreme value distribution (Gumbel, 1935) to annual maximum storage deficits. Catchment $S_{r,MM}$ is estimated as a weighted sum of the high and low vegetation $S_r$, based on the coverage fraction of high ($C_H$) and low ($C_L$) vegetation in the corresponding grid cell of that specific catchment, described by:

$$S_{r,MM} = C_L \, S_{r,L,2yr} + C_H \, S_{r,H,40yr} \tag{7}$$

## 2.4 HTESSEL model description

In this study we use the Hydrology Tiled ECMWF Scheme for Surface Exchanges over Land (HTESSEL) land surface model (Balsamo et al., 2009). This section presents the model parameterization of vegetated areas in the HTESSEL control model version (hereinafter CTR) based on the IFS documentation of cycle CY43R1 and the model codes itself (ECMWF, 2016). The core structure of this model is described by van den Hurk et al. (2000) and major changes in the hydrology parameterization were made by Balsamo et al. (2009) with the implementation of a global soil texture map instead of a single soil type, and a

runoff scheme accounting for subgrid variability, which resulted in improvements in global water budget simulations (Balsamo et al., 2011).

Figure 4a represents a simplified 3D view of a single grid cell. The HTESSEL model describes eight different surface fractions within a grid cell (ECMWF, 2016), but we only considered the vegetation covered fractions (high and low vegetation) because of the presence of roots. Considering exclusively vegetated areas, the grid cell surface is subdivided into high and low

vegetation covered area ($C_H$ and $C_L$) with a dominant type of vegetation ($T_H$ and $T_L$) based on the GLCC1.2 vegetation database. This database distinguishes 18 different vegetation types (e.g. evergreen broadleaf; tall grass; crops), each described with vegetation specific parameters based on experiments and literature (e.g. minimum canopy resistance; root distribution). The subsurface has a single soil texture based on FAO (2003) and is subdivided into four model layers with a total depth $z$ of 2.89 m that is kept uniformly constant in the global domain.

Figure 4b presents the connection of the subsurface with the surface, through roots and transpiration fluxes ($E_t$) in more detail. $S_r$ is not explicitly described in the model parameterization and, therefore, it is formulated based on our own understanding of its relation to the HTESSEL vegetation and root zone parameterizations (Eq. 8). Vegetation has roots in all four model soil layers (except for vegetation types desert and tundra that can only access the upper layer and the upper three layers, respectively (ECMWF, 2016)). There is a variable root distribution across the layers that is different for each vegetation type.

The vegetation specific root distribution ($R_k$) describes the root fraction with respect to the total amount of roots in each model soil layer. At a single time step, the capability of roots to extract soil moisture ($\theta_{k,roots}$, represented by the brown boxes in 4b) is a function of $R_k$ and the layer unfrozen soil moisture content ($\theta_k$). Thus, the more roots we have in a soil layer, the more moisture can be extracted at each time step. In the long term, however, the vegetation is able to extract all the plant available




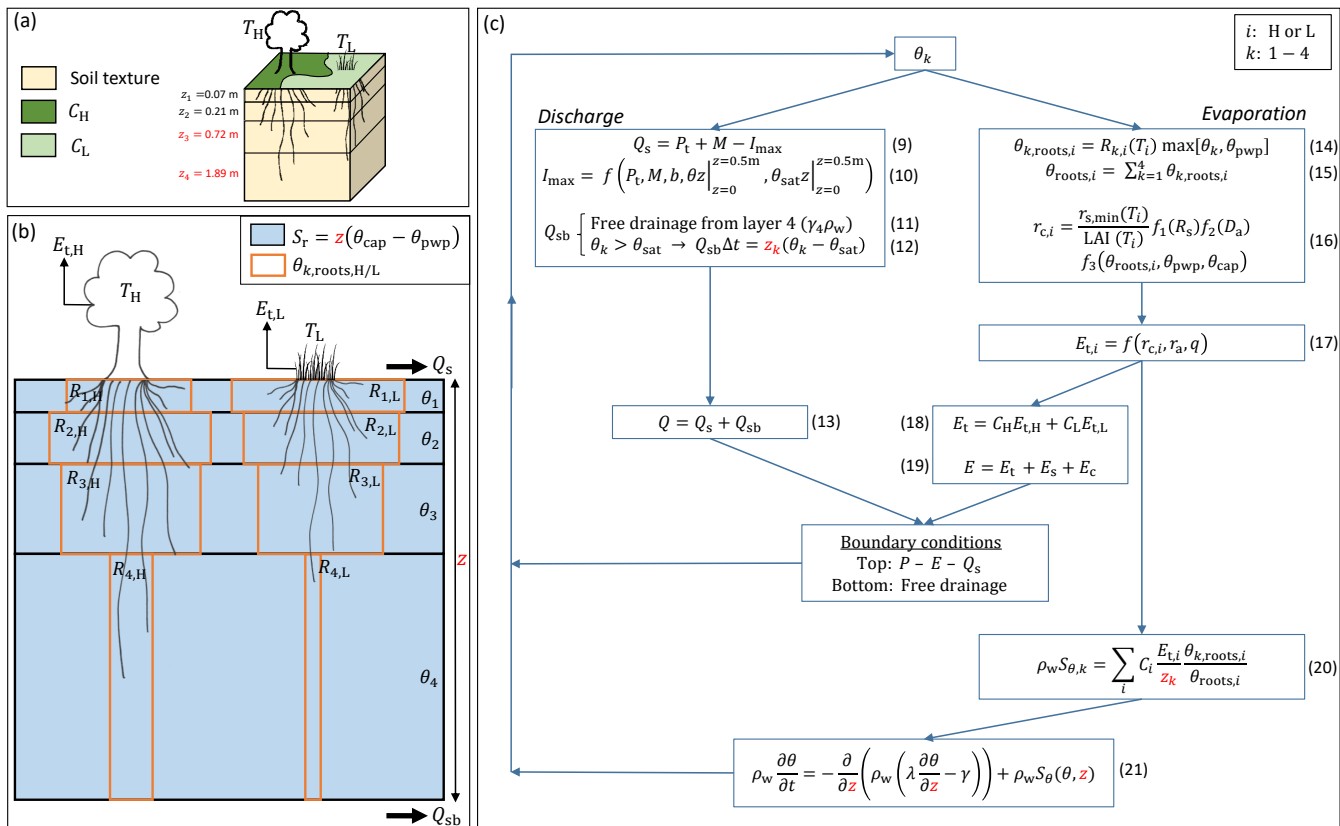

**Figure 4.** Root zone parameterization in the HTESSEL CTR version with highlighted in red the directly changed parameters in the HTESSEL MD version. (a) 3D overview of a single grid cell. (b) Schematic image of the four layer subsurface. (c) Scheme of equations for the calculation soil moisture, discharge and evaporation. The symbols in this figure are as follows, with $i$ high (H) and low (L) vegetation and $k$ layers 1–4: $C$ (−) vegetation coverage, $T$ dominant vegetation type, $z$ (m) layer depth, $P$ (m s$^{-1}$) precipitation, $P_t$ (m s$^{-1}$) precipitation through-fall, $M$ (m s$^{-1}$) snow-melt, $Q$ (m s$^{-1}$) total discharge, $Q_s$ (m s$^{-1}$) surface runoff, $Q_{sb}$ (m s$^{-1}$) subsurface runoff, $I_{max}$ (m s$^{-1}$) maximum infiltration rate, $b$ (−) variable representing sub-grid orography, $E$ (m s$^{-1}$) total evaporation, $E_t$ (m s$^{-1}$) transpiration, $E_s$ (m s$^{-1}$) soil evaporation, $E_c$ (m s$^{-1}$) canopy evaporation, $R$ (%) root distribution, $\theta$ (m$^3$m$^{-3}$) unfrozen soil moisture, $\theta_{pwp}$ (m$^3$m$^{-3}$) soil moisture at permanent wilting point, $\theta_{cap}$ (m$^3$m$^{-3}$) soil moisture at field capacity, $\theta_{sat}$ (m$^3$m$^{-3}$) soil moisture at saturation, $S_r$ (m) the root zone storage capacity, $\theta_{roots}$ (m$^3$m$^{-3}$) the root extraction efficiency, $r_c$ (s m$^{-1}$) canopy resistance, $r_a$ (s m$^{-1}$) atmospheric resistance, $R_s$ (W m$^{-2}$) downward shortwave radiation, $D_a$ (hPa) atmospheric water vapour deficit, $q$ specific humidity (kg kg$^{-1}$), $r_{s,min}$ (s m$^{-1}$) minimum canopy resistance, LAI (−) Leaf Area Index, $S_\theta$ (m$^3$m$^{-3}$s$^{-1}$) root extraction rate, $\gamma$ (m s$^{-1}$) hydraulic conductivity, $\lambda$ (m$^2$s$^{-1}$) hydraulic diffusivity and $\rho_w$ (kg m$^{-3}$) density of water.

soil moisture in the layers where roots are present. Therefore, $S_{r,CTR}$, represented in blue in Fig. 4b, is described by:

$\quad S_{r,CTR} = z(\theta_{cap} - \theta_{pwp})$ (8)





with $z$ the combined depth of all soil layers with roots ($z = 2.89$ m is a default value in HTESSEL for all vegetation types except for desert and tundra), and $\theta_{\mathrm{cap}} - \theta_{\mathrm{pwp}}$ the plant available moisture which is constant over the four soil layers. The plant available moisture is bounded by the soil texture specific moisture contents at field capacity ($\theta_{\mathrm{cap}}$), above which soil moisture drains by gravity, and at wilting point ($\theta_{\mathrm{pwp}}$), below which soil moisture is not accessible to roots. Runoff from the system

occurs as surface runoff ($Q_{\mathrm{s}}$) and subsurface runoff ($Q_{\mathrm{sb}}$).

Figure 4c presents the equation scheme of HTESSEL for calculating soil moisture and discharge and evaporation fluxes, with $i$ high (H) or low vegetation cover (L) and $k$ the four soil layers. The relative soil moisture content $\theta$ controls the calculations of discharge and evaporation fluxes. The surface runoff ($Q_{\mathrm{s}}$) is defined by the precipitation through-fall ($P_{\mathrm{t}}$), snow-melt ($M$) and the maximum infiltration rate ($I_{\mathrm{max}}$) (Eq. 9). $I_{\mathrm{max}}$ is a function of $P_{\mathrm{t}}$, $M$, a spatially variable parameter ($b$), that is defined by the

standard deviation in sub-grid orography, and the vertically integrated (top 0.5 m) soil moisture ($\theta$) and saturation soil moisture ($\theta_{\mathrm{sat}}$) (Eq. 10) (Dümenil and Todini, 1992; van den Hurk and Viterbo, 2003). The subsurface runoff ($Q_{\mathrm{sb}}$) consists of two components: free drainage from layer 4, that is a function of hydraulic conductivity in this layer ($\gamma_4$) and water density ($\rho_{\mathrm{w}}$) (Eq. 11) and the excess absolute soil moisture when $\theta_k > \theta_{\mathrm{sat}}$ (Eq. 12). Total discharge ($Q$) is the sum of $Q_{\mathrm{s}}$ and $Q_{\mathrm{sb}}$ (Eq. 13). The average root extraction efficiency in all layers ($\theta_{\mathrm{roots}}$) is described by Eq. 14 and Eq. 15 as the weighted sum of the vegetation

specific $R_k$ and $\theta_k$. The canopy resistance ($r_{\mathrm{c}}$) (Eq. 16) describes the resistance of vegetation to transpiration and is a function of vegetation specific values for minimum canopy resistance ($r_{\mathrm{s,min}}$) and LAI, a function of shortwave radiation ($f_1(R_{\mathrm{s}})$), a function of atmospheric water vapour deficit ($f_2(D_{\mathrm{a}})$) and a function of the root extraction efficiency ($f_3(\theta_{\mathrm{roots},i}, \theta_{\mathrm{pwp}}, \theta_{\mathrm{cap}})$. The canopy resistance defines $E_{\mathrm{t,i}}$ together with specific humidity ($q$) and an atmospheric resistance term ($r_{\mathrm{a}}$) (Eq. 17). Total $E_{\mathrm{t}}$ is a weighted sum of the separate transpiration products based on the sub-grid coverage $C_{\mathrm{L}}$ and $C_{\mathrm{H}}$ (Eq. 18) and total

evaporation ($E$) is a sum of transpiration ($E_{\mathrm{t}}$), soil evaporation ($E_{\mathrm{s}}$) and canopy evaporation ($E_{\mathrm{c}}$) fluxes (Eq. 19). The detailed formulations of the latter two fluxes are not relevant in this study and, therefore, not included in this model description. $E_{\mathrm{t,i}}$ is attributed to the different soil layers in the calculation of the root extraction ($S_\theta$) based on the layer depth ($z_k$) and $\theta_{k,\mathrm{roots}}$ (Eq. 20). The change in soil moisture over time ($\partial\theta/\partial t$) is calculated applying the Darcy-Richards equation with $\gamma$ and $\lambda$ hydraulic conductivity and diffusivity (Eq. 21). This equation is solved with top soil boundary condition of $P - E - Q_{\mathrm{s}}$, and a bottom

soil boundary condition of free drainage.

### 2.5    Implementation of memory method root zone storage capacity estimates in HTESSEL

Here we develop an approach to implement the climate controlled $S_{\mathrm{r,MM}}$ (results in Sect. 3.1) in HTESSEL, while maintaining the modeling framework of the CTR model described in Sect. 2.4. We found that $S_{\mathrm{r,CTR}}$ is exclusively defined by the soil type and the model soil depth ($z$) (Eq. 8). In our modified version of HTESSEL, hereafter referred to as the Moisture Depth (MD)

model, the soil depth for moisture calculations is changed to satisfy the following equation:

$$S_{\mathrm{r,MD}} = S_{\mathrm{r,MM}} = z_{\mathrm{MD}}(\theta_{\mathrm{cap}} - \theta_{\mathrm{pwp}}) \tag{22}$$

This depth change is achieved by changing model layer 4, except in the case this would lead the model depth of layer 4 to approach zero ($z_4 \approx 0$). In this case a minimum threshold (0.2 m) is set for $z_4$ and the depth of layer 3 is further changed





to obtain $S_{r,MD} = S_{r,MM}$ as required in Eq. 22. This is necessary because $z_4 \approx 0$ in the moisture calculation would cause
inconsistencies in the thermal diffusion calculations as the layer soil temperature is a function of the layer soil moisture.
It should be noted that the layer depths for thermal diffusion calculations are not modified in the MD model. The directly
changed parameters in MD are highlighted in red in Fig. 4.

## 2.6    Model simulations

Simulations are performed in a standalone version of HTESSEL (Balsamo et al., 2009) as it was implemented in the frame of
version 3 of the EC-EARTH Earth system model (http://www.ec-earth.org) for both the CTR (Sect. 2.4) and MD (Sect. 2.5)
model versions. The model is forced with 3-hourly GSWP-3 atmospheric boundary conditions (Kim, 2017) for the historical
time series 1970–2010, with the first five years used for spin-up. The spatial resolution of the HTESSEL model is a reduced
gaussian grid (N128), with the grid cells over Australia being approximately $5500\,\mathrm{km}^2$.

## 2.7    Model evaluation

Most study catchments are smaller than single HTESSEL grid cells (Table S1). For catchments completely falling within a
single HTESSEL grid cell, this cell is selected for analysis. In the case a catchment falls within more than one grid cell,
the average of the model output in the separate grid cells is used for analysis. The model performances of CTR and MD
are compared based on modeled monthly discharge and evaporation fluxes for 1975–2010: long-term annual means, monthly
seasonal climatology and inter-annual anomalies of monthly fluxes (monthly fluxes minus monthly climatology) are evaluated.
Modeled $Q$ is compared to station observations and modeled $E$ to the FLUXCOM-WB evaporation (Sect. 2.2 and Eq. 1).
For long-term annual means, the percent-bias between the reference and modeled fluxes is calculated (evaporation p-bias $=$
$(\overline{E}_{mod} - \overline{E}_{ref})/\overline{E}_{ref}$). For the monthly seasonal climatology and inter-annual anomalies, the model performance is quantified
by using the Pearson correlation coefficient ($r$) and a variability performance metric ($v = (1 - \alpha)^2$) that depends on the ratio
of modeled and reference standard deviation ($\alpha = \sigma_{mod}/\sigma_{ref}$). These performance metrics are calculated for the individual
catchments, and then averaged to evaluate model performance over tropical, temperate and Mediterranean climate regions.

To test significance of the improvement in model performance of MD compared to CTR, a Monte Carlo bootstrap method
(1000 repetitions) is employed. The 1000 samples are taken by resampling randomly with replacement among CTR and MD
values at each time-step. The null hypothesis of getting as high or higher performance parameters simply by chance is tested
at the 5% and 10% significance levels, for the individual catchments as well as for the performance averages over the tropical,
temperate and Mediterranean climate regions. P-values of the model improvements are provided in the Supplementary Material
(Tables S5 and S6).



# 3 Results

## 3.1 Root zone storage capacity estimates

Figure 5 shows that there is no relation between $S_{r,MM}$ and $S_{r,CTR}$. The range of $S_{r,MM}$ (125–722 mm) in the study catchments

is much larger than the range of $S_{r,CTR}$ (491–725 mm), indicating that HTESSEL may not adequately represent the spatial heterogeneity of $S_r$ (Table S2). The range of $S_{r,MM}$ in the catchments is consistent with Wang-Erlandsson et al. (2016), who found similar ranges of $S_r$ (approximately 100–600 mm) over Australia by using gridded products of $S_r$ based on rooting depths from observations and optimised inverse modelling, and global $S_{r,MM}$ estimated based on satellite evaporation products. $S_{r,MM}$ estimates are on average smaller in the five temperate (194 mm) catchments than in the three Mediterranean (321 mm)

and the seven tropical (437 mm) catchments. In the tropical and Mediterranean regions vegetation needs to bridge extensive dry seasons as rainfall seasonality is high (Fig. 2, Table 1), resulting in larger $S_{r,MM}$ than in temperate regions with year-round precipitation. In the Mediterranean, the average time-lag between $P$ and $E_p$ of 5.7 months results in large root zone storage deficits in the hot and dry summers, and therefore, larger $S_{r,MM}$ than in the temperate catchments.

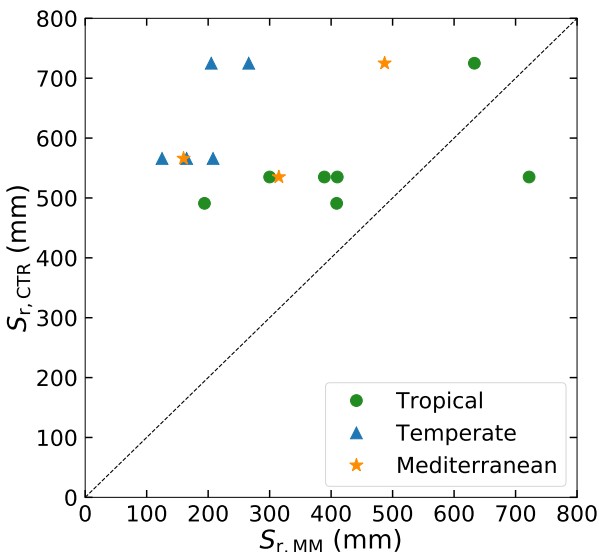

**Figure 5.** Catchment $S_r$ as estimated from the memory method ($S_{r,MM}$) compared to the HTESSEL CTR parameterization ($S_{r,CTR}$) in the catchment corresponding grid-cells.

## 3.2 Long-term mean annual climatology

The HTESSEL CTR version overestimates observed $\overline{Q}$ in 9 out of 15 catchments with on average $40\,\mathrm{mm\,year^{-1}}$ (tropical), $3\,\mathrm{mm\,year^{-1}}$ (temperate) and $122\,\mathrm{mm\,year^{-1}}$ (Mediterranean) (Table 2; Table S3; Table S4). This overestimation of observed $Q$ goes together with an average underestimation of $\overline{E}_{WB}$ by CTR. As $S_{r,MM}$ is generally smaller than $S_{r,CTR}$ (Fig. 5), the MD




version results in reduced $\overline{E}$ and increased $\overline{Q}$ compared to CTR, but the changes are quite small (Table 2). The MD increase
in modeled $\overline{Q}$ compared to CTR results on average in larger p-biases in tropical (+16.9 % vs. +13.7 %), temperate (+24.4 %
vs. +4.9 %) and Mediterranean (+263.8 % vs. +249.9 %) catchments, but the results are largely variable among the individual
catchments (Table S4).

**Table 2.** Long-term annual mean modeled discharge ($\overline{Q}$) and evaporation ($\overline{E}$) in the HTESSEL CTR and MD versions for the tropical,
temperate and Mediterranean climate regions (catchment averages) and reference $\overline{Q}$ (station observations) and $\overline{E}$ ($\overline{E}_{WB}$ (Sect. 2.2)). The
p-biases of the modeled climate region average $\overline{Q}$ and $\overline{E}$ are presented between brackets. Similar values for the individual catchments are
shown in Tables S3 and S4.

| Climate region | $\overline{Q}$ (mm year$^{-1}$) | | | $\overline{E}$ (mm year$^{-1}$) | | |
|---|---|---|---|---|---|---|
|  | Observations | HTESSEL CTR | HTESSEL MD | WB | HTESSEL CTR | HTESSEL MD |
| Tropical | 291 | 331 (+13.7%) | 340 (+16.9%) | 834 | 790 (-5.3%) | 781 (-6.5%) |
| Temperate | 56 | 59 (+4.9%) | 70 (+24.4%) | 626 | 624 (-0.4%) | 611 (-2.4%) |
| Mediterranean | 49 | 171 (+249.9%) | 177 (+263.8%) | 836 | 717 (-14.2%) | 709 (-15.2%) |

## 3.3 Monthly seasonal climatology

Although $\overline{Q}$ does not considerably change in MD compared to CTR (Sect. 3.2), MD reproduces the seasonal variations in $Q$
considerably better than CTR (Fig. 6a–c and Table 3). In the tropical and Mediterranean catchments, MD increases $Q$ in the
wet months while it decreases $Q$ in the dry months compared to CTR, and hence improves the seasonal timing of observed
$Q$ (Fig. 6a,c and Table 3). In the temperate catchments, MD increases $Q$ in the wet months (Jul–Sep) compared to CTR in
accordance with observations, although in the other months the changes of MD compared to CTR are mixed (Fig. 6b). In terms
of the correlation between modeled and observed monthly seasonal climatology, $Q$ improved in MD compared to CTR in 12
out of 15 catchments, with 7 catchments passing the 5% significance level for improvement (Table S5). For the climate region
averages, the correlation significantly improved in MD from 0.84 to 0.90 (tropical), from 0.74 to 0.86 (temperate) and from
0.86 to 0.96 (Mediterranean) compared to CTR (Table 3). On average, MD resulted in larger variations in monthly $Q$ than CTR
(Fig. 6a–c). The variability term $v = (1 - \sigma_{mod}/\sigma_{obs})^2$ improved from 0.17 to 0.06 (tropical) and from 0.17 to 0.10 (temperate)
in MD compared to CTR, but in the Mediterranean catchments the models strongly overestimate the observed variations in $Q$
(Fig. 6c) with the variability term increasing from 2.80 in CTR to 8.73 in MD (Table 3; Table S5).

In contrast to the improvement in monthly seasonal climatology of $Q$ in MD, the monthly seasonal cycle of $E$ appears to be
not much affected as reported in Fig. 6d–f and Table 3.





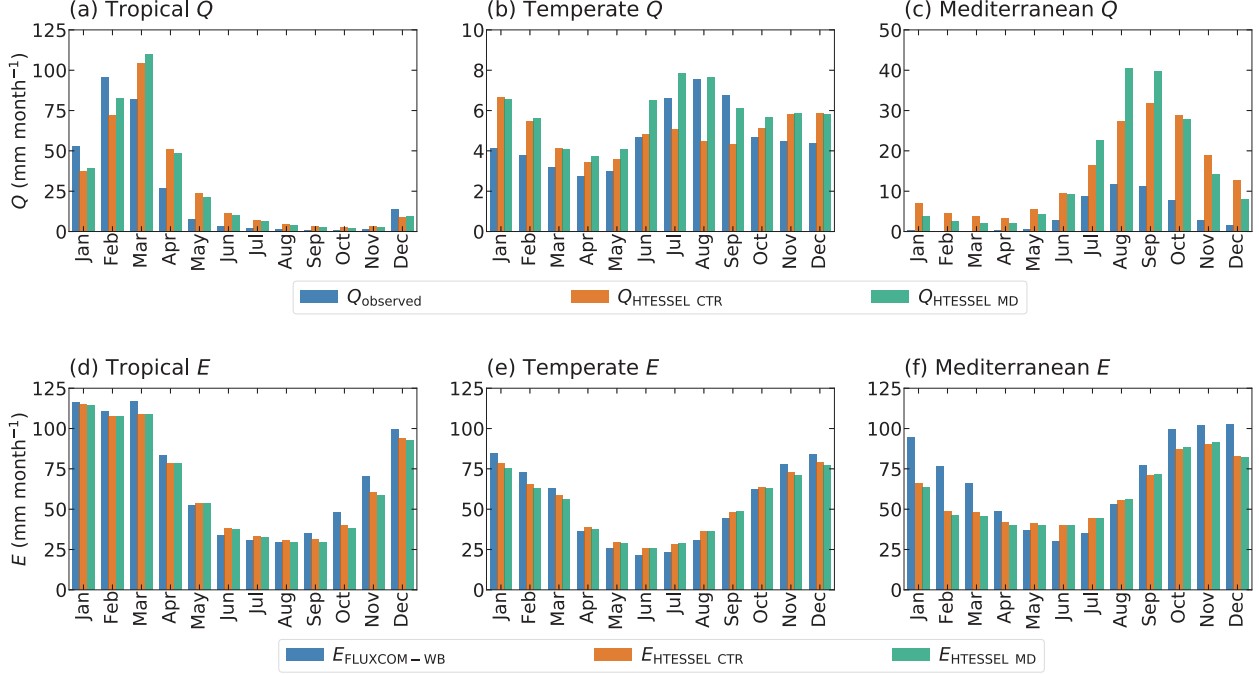

**Figure 6.** Monthly seasonal climatology of observed discharge ($Q$) (top) and FLUXCOM-WB evaporation ($E_{\text{FLUXCOM-WB}}$) (bottom) and modeled values in the HTESSEL CTR and MD versions, averaged for the tropical (a, d), temperate (b, e) and Mediterranean (c, f) catchments for the time series 1975–2010. Similar figures for the individual catchments are shown in Fig. S1 ($Q$) and Fig. S2 ($E$).

### 3.4 Inter-annual monthly anomalies

Figure 7a and 7c show that MD is better in capturing the variations in inter-annual $Q$ anomalies than CTR in the presented tropical and temperate catchments, while in the Mediterranean catchment both models strongly overestimate the inter-annual

$Q$ anomalies compared to observations (Fig. 7e). In 14 out of 15 catchments, the variability in the inter-annual $Q$ anomalies increases in MD compared to CTR (Fig. S1; Table S5). This results in an average improvement in the inter-annual anomaly variability ($v$) from 0.12 to 0.11 (tropical) and from 0.09 to 0.06 (temperate) in MD compared to CTR (Table 4). However, in the Mediterranean catchments, the increased variability in the $Q$ anomalies leads to a strong overestimation of $Q$ anomalies with respect to observations (Fig. 7e; Fig. S1m–o), with $v$ increasing from 0.99 in CTR to 4.26 in MD. Figures 7a, 7c and 7e

also show that the timing of the $Q$ anomalies improves in MD compared to CTR, with in particular the improved timing of the falling limbs clearly visible in Fig. 7a and 7e. The inter-annual $Q$ anomaly correlation (corresponding to the timing) improves in 14 out of 15 catchments, with 9 catchments passing the 5% significance level for improvement (Table S5). On average, the correlation ($r$) increases from 0.74 to 0.78 (tropical), from 0.80 to 0.85 (temperate) and from 0.71 to 0.79 (Mediterranean) in MD compared to CTR. In contrast to the improvement in the inter-annual $Q$ anomalies in MD, the inter-annual $E$ anomalies

do not considerably change compared to CTR (Fig. 7b,d,f; Table 4, Table S6).





**Table 3.** Model performance parameters of monthly seasonal discharge ($Q$) and evaporation ($E$) climatologies (1975–2010), with $r$ representing pearson correlation and $v = (1 - \alpha)^2$ variability, with $\alpha = \sigma_{\mathrm{mod}}/\sigma_{\mathrm{obs}}$, in tropical, temperate and Mediterranean climate regions for the HTESSEL CTR and MD versions (catchment averages). Modeled $Q$ is compared to station observations and modeled $E$ to FLUXCOM-WB (Eq. 1). For $r$, a value of 1 represents a perfect model, for $v$ a value of 0 represents a perfect model. The significance test of the MD improvements compared to CTR is represented by [**] (passing 5% level) and [*] (passing 10% level). Values of $r$ and $\alpha$ for the individual catchments and p-values of improvement are shown in Tables S5 ($Q$) and S6 ($E$).

| | | Discharge | | Evaporation | |
|---|---|---|---|---|---|
| Climate region | HTESSEL version | $r$ (-) | $v$ (-) | $r$ (-) | $v$ (-) |
| Tropical | CTR | 0.84 | 0.17 | 0.98 | 0.07 |
| | MD | 0.90[**] | 0.06[**] | 0.98 | 0.07 |
| Temperate | CTR | 0.74 | 0.17 | 0.99 | 0.04 |
| | MD | 0.86[**] | 0.10[**] | 0.98 | 0.05 |
| Mediterranean | CTR | 0.86 | 2.80 | 0.81 | 0.08 |
| | MD | 0.96[*] | 8.73 | 0.80 | 0.07 |

**Table 4.** Model performance parameters of inter-annual monthly discharge ($Q$) and evaporation ($E$) anomalies (1975–2010), with $r$ representing pearson correlation and $v = (1 - \alpha)^2$ variability, with $\alpha = \sigma_{\mathrm{mod}}/\sigma_{\mathrm{obs}}$, in tropical, temperate and Mediterranean climate regions for the HTESSEL CTR and MD versions (catchment averages). Modeled $Q$ is compared to station observations and modeled $E$ to FLUXCOM-WB (Eq. 1). For $r$, a value of 1 represents a perfect model, for $v$ a value of 0 represents a perfect model. The significance test of the MD improvements compared to CTR is represented by [**] (passing 5% level) and [*] (passing 10% level). Values of $r$ and $\alpha$ for the individual catchments and p-values of improvement are shown in Tables S5 ($Q$) and S6 ($E$).

| | | Discharge | | Evaporation | |
|---|---|---|---|---|---|
| Climate region | HTESSEL version | $r$ (-) | $v$ (-) | $r$ (-) | $v$ (-) |
| Tropical | CTR | 0.74 | 0.12 | 0.79 | 1.39 |
| | MD | 0.78[**] | 0.11 | 0.79 | 1.48 |
| Temperate | CTR | 0.80 | 0.09 | 0.81 | 1.12 |
| | MD | 0.85[**] | 0.06[*] | 0.82[**] | 1.46 |
| Mediterranean | CTR | 0.71 | 0.99 | 0.78 | 1.17 |
| | MD | 0.79[**] | 4.26 | 0.78 | 1.31 |





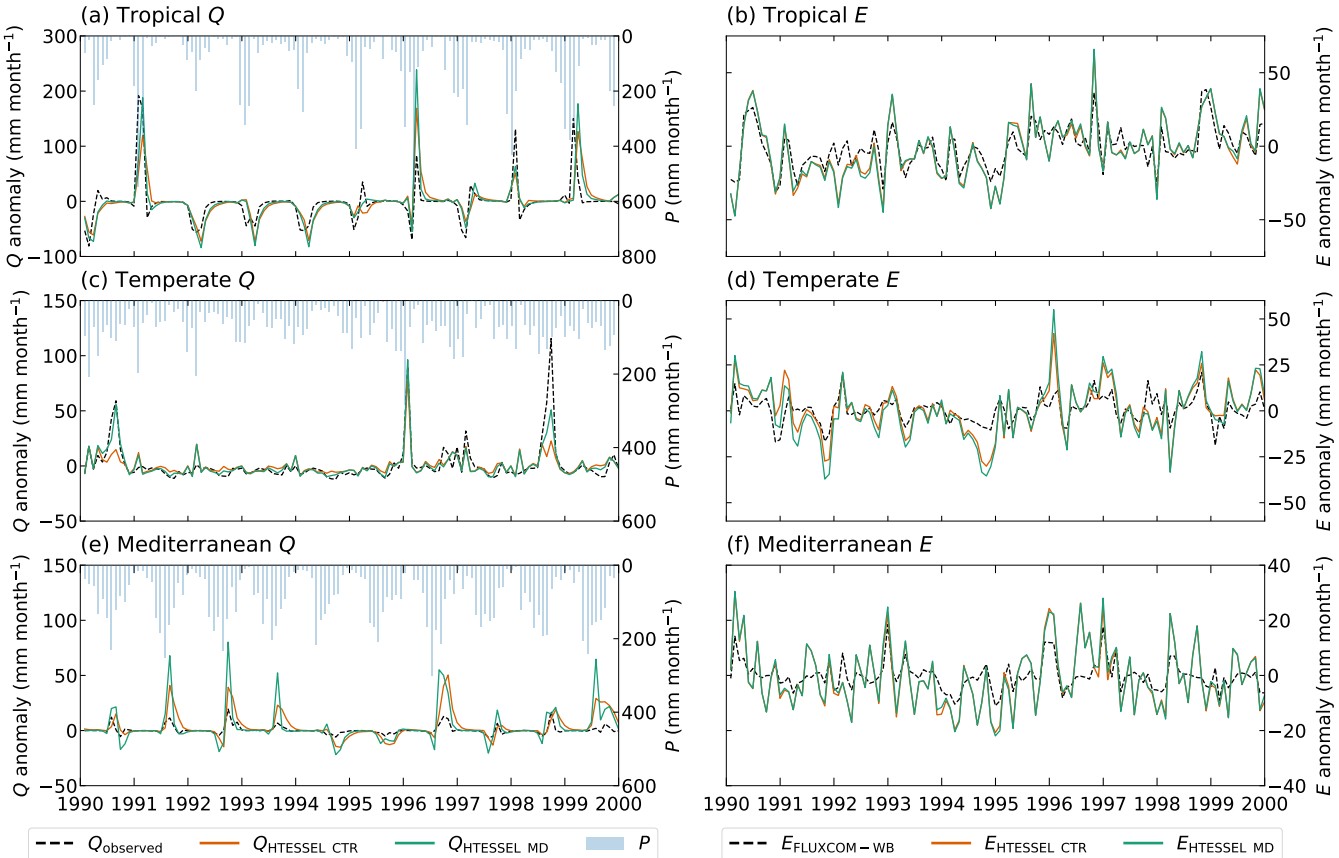

**Figure 7.** Inter-annual monthly anomalies of observed discharge ($Q$) (left) and FLUXCOM-WB evaporation ($E$) (right) fluxes and modeled values in the HTESSEL CTR and MD versions, in an individual representative tropical (catchment Mi) (a, b), temperate (catchment Na) (c, d) and Mediterranean (catchment K) (e, f) catchment based on the time series 1975–2010. Similar figures for the individual catchments are shown in Fig. S1 ($Q$) and Fig. S2 ($E$).

## 4 Discussion

### 4.1 Synthesis of results

$S_{r,MM}$ is lower than $S_{r,CTR}$ in 14 out of 15 catchments (Fig. 5). This is seemingly in contrast with literature suggesting that the root depth in land surface models is too low and that the absence of deep roots is a cause for uncertainties in simulated

evaporation (Kleidon and Heimann, 1998; Pan et al., 2020; Sakschewski et al., 2020). However, $S_r$ represents the water volume accessible to roots and is therefore not necessarily proportional to root depth as a small $S_r$ does not preclude the presence of deep roots, as illustrated in Fig. 4 in Singh et al. (2020).

The modeling results show that the difference in long-term mean $\overline{Q}$ and $\overline{E}$ fluxes between CTR and MD are small (Table 2), whereas the differences between monthly (climatological and inter-annual) variations are clearly visible (Fig. 6 and Fig.





7). This corresponds to other studies on catchment hydrology that suggest that the root zone storage mainly affects the fast hydrological response of a catchment (Oudin et al., 2004; Euser et al., 2015; Nijzink et al., 2016; De Boer-Euser et al., 2016). Furthermore, previous studies found larger improvements of modeled discharge using $S_{r,MM}$ in humid regions with large rainfall seasonality (De Boer-Euser et al., 2016; Wang-Erlandsson et al., 2016). This is not found in our study, as we obtain slightly smaller improvements in the discharge correlation for the tropical catchments than for the temperate and Mediterranean ones.

This is at least partly related to the smaller difference between $S_{r,MM}$ and $S_{r,CTR}$ in the tropical catchments than in temperate and Mediterranean ones (Fig. 5). The Mediterranean catchments have large climatological biases and too large discharge variability in the seasonal cycle and inter-annual anomalies in CTR, and MD further degrades the performance with respect to bias and variability (Tables 2, 3 and 4). On the other hand, the correlation of seasonal climatology and inter-annual anomalies consistently improves in all climate regions with the implementation of $S_{r,MM}$. Therefore, it is suggested that other aspects of

the hydrology parameterization than $S_r$ could be primarily leading to the large climatological biases and too large discharge variability in the seasonal cycle and inter-annual anomalies in the Mediterranean. On the other hand, uncertainties in the GSWP-3 forcing could also in part cause the large biases in the Mediterranean. In this climate region, it is found that GSWP-3 $\overline{P}$ (0.5°grid) is considerably larger than $\overline{P}$ from the SILO dataset, that provides $P$ on a 0.05°grid directly derived from ground-based observational data (Jeffrey et al., 2001).

Although we found significant differences in modeled $Q$ between CTR and MD, the discrepancy in $E$ was very limited in all climate regions (Table 3; Table 4; Table S6; Fig. S2). This lack of evaporation sensitivity to $S_r$ is unexpected and requires more in depth evaluation of HTESSEL. In this respect, it would be useful to check how this insensitivity of evaporation to $S_r$ changes is model dependent and compare HTESSEL behavior with other LSMs in a multi-model context, comparable to for example van den Hurk et al. (2016) and Ardilouze et al. (2017). On the other hand, we also expect this to be related to methodology

applied which will be further discussed in Section 4.3.

## 4.2 Methodological uncertainty

Although the catchments were selected carefully, their location and sizes do not completely match with the HTESSEL grid cells. Thus, assuming a one to one relation between precipitation, evaporation, river discharge and root zone storage capacities at the catchment and the grid cell is a potential source of error. However, this configures as random error and is therefore likely

to cancel out in multiple catchment settings as is done in this study. Another source of uncertainty is the parameterization of the memory method for estimating catchment $S_r$. This method requires estimations of maximum interception storage, seasonal and inter-annual transpiration signals and return periods, which lead to differences in $S_{r,MM}$ when other values are chosen. A sensitivity analysis of $S_{r,MM}$ with a high $S_{r,MM}$ ($S_{i,max} = 1.5\,\mathrm{mm}$, $RP_{low} = 3\,\mathrm{years}$, $RP_{high} = 60\,\mathrm{years}$, $f = 0.15$ (see Appendix A)) and a low $S_{r,MM}$ ($S_{i,max} = 3.5\,\mathrm{mm}$, $RP_{low} = 1.5\,\mathrm{years}$, $RP_{high} = 20\,\mathrm{years}$, $f = 0.35$) on average deviated $45\,\mathrm{mm}$ from the

average $S_{r,MM}$ estimates used in this study ($S_{i,max} = 2.5\,\mathrm{mm}$, $RP_{low} = 2\,\mathrm{years}$, $RP_{high} = 40\,\mathrm{years}$, $f = 0.25$). This deviation is small considering the average $S_{r,MM}$ being $319\,\mathrm{mm}$. Besides, irrigation, as a possible external water source in catchments with crops (Table S1), and deep groundwater, as a water source for deep-rooting vegetation, are not accounted for in the approach. However, we think that the estimation of transpiration is the main uncertainty in the approach. The assumption that the seasonal




variations in $E_t$ and $E_p$ are in phase may not hold in Mediterranean regions where $E_p$ and $P$, and thereby the water available for
transpiration, tend to be out of phase. Applying the seasonal pattern of transpiration modeled by CTR to the memory method
in Mediterranean catchments results in smaller $S_{r,MM}$ estimates in these catchments (average: 292 mm) than with the initial
approach where the seasonality of $E_t$ was based on $E_p$ (average: 321 mm). The relatively low deviation for both the parameter
uncertainty and the uncertainty in the timing of $E_t$ leads us to conclude that these assumptions have a small impact on the
general finding that $S_{r,MM}$ is lower than $S_{r,CTR}$ and that HTESSEL does not represent the spatial heterogeneity of $S_r$.

Station observations of river discharge are used in both the $S_{r,MM}$ estimation and the model evaluation. However, because
the memory method is only based on observations of long term annual mean discharge ($\overline{Q}$) and the model evaluation is mainly
based on the monthly seasonal and inter-annual variations in $Q$, we consider model evaluation based on these data appropriate.

### 4.3   Root zone storage capacity implementation

The HTESSEL CTR version does not explicitly formulate $S_r$ and, therefore, we formulate $S_{r,CTR}$ based on the root zone
parameterization as presented in Sect. 2.4 in order to modify the model parameters in a way to make the model consistent
with the $S_{r,MM}$ estimates. This formulation represents the theoretical $S_{r,CTR}$, but it remains uncertain to what extent the soil
moisture in the four layers is actually used by the modeled vegetation, in particular in layer 4 because of the relatively small
root percentage prescribed from look-up tables in this layer for most vegetation types compared to the other layers. In MD the
depths for soil moisture calculations are changed, directly resulting in changes in absolute soil moisture and, thereby in indirect
changes in discharge and transpiration. This modification is relatively simple, flexible and there is no limitation in the possible
range of soil depths for moisture calculations and, therefore, could similarly be implemented in other land surface models.
However, it should be noted that this strategy chosen for changing the HTESSEL $S_r$ is not the only possible. As follows from
Eq. 8, also the plant available soil moisture ($\theta_{cap} - \theta_{pwp}$) defines the $S_r$. However, modifications in the model's $\theta_{cap}$ or $\theta_{pwp}$
are not desired as these parameters are soil texture specific properties. Moreover, modifications in the formulations of the root
available moisture for each time-step ($\theta_{roots}$) appears conceptually not meaningful.

      There are several alternative hypotheses that may potentially explain the limited sensitivity of modeled $E$ to the modified
$S_r$. First, the resistance of vegetation to transpiration is a function of the moisture supply (soil moisture) and the moisture
demand (atmospheric condition) (Eq. 14–16). The atmospheric conditions, that define moisture demand and thereby constrain
transpiration, are similar in both CTR and MD because the models are run in an offline version. Therefore, the soil moisture-
atmosphere feedback is not represented and the moisture demand side dominates the moisture supply side in the evaporation
calculations. This issue could be overcome by using coupled climate simulations. Second, although $S_r$ is changed in MD
compared to CTR, the parameterization of the vegetation water stress is kept constant. Ferguson et al. (2016) found that
different formulations of root water uptake considerably influence modeled water budgets and, therefore, it is likely that changes
in evaporation in MD compared to CTR are constrained by the vegetation water stress formulations (Eq. 14–16). Third, the
insensitivity of evaporation to the changes in model soil depth is probably also related to the fact that the resistance of vegetation
to transpiration is a function of the relative soil moisture ($\theta$), which is not directly affected by changing the soil depth. On the
other hand, soil depth changes directly affect the modeled $Q$, as modeled surface ($Q_s$) and subsurface runoff ($Q_{sb}$) directly



depend on the absolute moisture storage capacity of the soil (see Eq. 10 and Eq. 12), with $Q_s$ a function of the absolute moisture in the top $50\,\mathrm{cm}$ of soil and $Q_{sb}$ a function of the the absolute excess soil moisture when the layer's moisture content

exceeds saturation moisture content. Fourth, monthly fluxes of $Q$ are often a full order of magnitude smaller than $E$. Hence small changes in the partitioning simply add up to larger relative changes for $Q$.

## 5   Conclusions

This study is an attempt to overcome major limitations in the representation of the vegetation's root zone in land surface models. Specifically, we looked at the HTESSEL land surface model and found that the root zone storage capacity $S_r$ is only

as a function of soil texture and soil depth, the latter being kept constant over the modeled global domain (in HTESSEL $z = 2.89\,\mathrm{m}$), while from the state-of-the-art literature (e.g. Collins and Bras, 2007; Guswa, 2008; Gentine et al., 2012; Gao et al., 2014) it is indicated that $S_r$ is, to a large extent, climate controlled. We found that indeed the HTESSEL control version (CTR) does not adequately represent the spatial heterogeneity of $S_r$, with the range of $S_{r,CTR}$ (491–725 mm) much narrower than the range obtained for the climate controlled estimate $S_{r,MM}$ (125–722 mm) in 15 Australian catchments with contrasting

climate characteristics considered in this study. Furthermore, $S_{r,CTR}$ was found to be considerably larger than the climate controlled estimate $S_{r,MM}$ in 14 out of 15 catchments. We developed a new version of HTESSEL by suitably modifying the soil depths (MD) to obtain modeled $S_{r,MD}$ that matches $S_{r,MM}$ over the 15 catchments considered over Australia, while maintaining the overall HTESSEL model setup (Fig. 4). This strategy to modify the model's $S_r$ is relatively simple and could similarly be implemented in other land surface models. Moreover, the applied methodology would allow for a time-varying $S_r$ in LSMs,

and hence all four limitations of using sparse root observations mentioned in Sect. 1 could be overcome.

The comparison of the offline simulations with original (CTR) and modified (MD) versions of HTESSEL shows that the difference of the biases in modeled long-term mean climatology of discharge and evaporation fluxes is generally small. On the other hand, the seasonal timing of the discharge flux is significantly improved in MD indicating the beneficial effect of the climate controlled representation of $S_r$. Consistently, MD improves the correlation with observations for the monthly seasonal

climatology of discharge fluxes in 12 out of 15 catchments (with 7 catchments passing 5% significance level) and for the inter-annual monthly discharge anomalies in 14 out of 15 catchments (with 9 catchments passing 5% significance level) (Table S5). Considering the climate region averages, the correlations of monthly seasonal climatology significantly improve in MD compared to CTR from 0.843 to 0.902 (tropical), from 0.741 to 0.855 (temperate) and from 0.860 to 0.951 (Mediterranean). The averaged correlations of the inter-annual monthly anomalies significantly improve in MD compared to CTR from 0.741 to 0.778

(tropical), from 0.795 to 0.847 (temperate) and from 0.705 to 0.785 (Mediterranean). Surprisingly the modeled evaporation is shown to be relatively insensitive to changes in $S_r$. The apparent insensitivity of evaporation to the changes in model soil depth is probably mainly related to the fact that evaporation only depends on the relative moisture content in each soil layer, which in the model is not directly affected by the depth of the soil. On the other hand, surface and subsurface runoff depend on the cumulative moisture content of the soil at any given time. Other than the relative moisture content this depends on the

absolute moisture storage capacity of the soil that will vary together with the change in soil depth. Moreover, small changes





in absolute fluxes translated to larger relative changes for runoff compared to evaporation (Fig. 6). As a final conclusion, we
believe that a global application of climate controlled root zone parameters has the potential to improve the timing of modeled
water fluxes by land surface models, but from the results of this study a significant reduction of annual-mean climatological
biases cannot be expected. More work will be needed in the future to improve long-term mean simulation of discharge and
evaporation fluxes by exploiting station-based and latest-generation satellite observations. To this aim the use of coordinated
multi-model frameworks for the intercomparison of state-of-the-art LSMs could be fundamental.

*Code and data availability.* Catchment discharge observations were taken from the Australian Bureau of Meteorology and can be down-
loaded from http://www.bom.gov.au/water/hrs/. FLUXCOM evaporation data were taken from the FLUXCOM initiative and can be down-
loaded from http://www.fluxcom.org/EF-Download/. Top of the atmosphere radiation data were taken from Mines ParisTech and can be
downloaded from http://www.soda-pro.com/web-services/radiation/extraterrestrial-irradiance-and-toa/. The offline HTESSEL model was
provided by EC-EARTH, together with the GSWP-3 forcing data, vegetation and soil data. The adapted modules, model output and analysis
codes are available upon request. The python scripts used for $S_r$ calculation and statistical significance of the results can be downloaded from
https://github.com/fvanoorschot/Python-scripts-van-Oorschot-2021/.

## Appendix A: Iterative procedure for transpiration estimation

Daily transpiration is estimated by Eq. (6) with $c$ a coefficient that represents the ratio between transpiration and potential
evaporation (Sect. 2.3). With $c = \overline{E_t}/\overline{E_p}$ as a constant value, we do not account for inter-annual variability in transpiration
caused by the interplay between atmospheric water demand and vegetation-available water supply. Therefore, we add an
iterative procedure to estimate annually varying values for $c$, which is described here.

Steps 1 to 6 describe the procedure used to estimate $c$ with step 1 the initial estimates and step 2 to 6 executed iteratively. $i$
represents the iterations (0–9) and $a$ the hydrological years (1973–2010). $P_e$, $E_t$, $E_p$ and $S_d$ are daily values. After ten iterations
($i = 9$) the resulting annual transpiration estimates stabilized and the corresponding storage deficits were used for the Gumbel
$S_r$ analysis as described in Sect. 2.3.

1. Initial estimates ($i = 0$) of $E_t$ and $S_d$ with a constant $c_{0,a} = \overline{E_t}/\overline{E_p}$ for $a = 1973$–2010.

$$E_{t,0}(t) = c_{0,a} E_p(t) \tag{A1}$$


$$S_{d,0} = \max(0, -\int_{1973}^{2010} (P_e - E_{t,0})\mathrm{d}t) \tag{A2}$$

2. Calculate the annual change in storage in the root zone ($S$) with $t_0$ and $t_1$ the start and end of a hydrological year.

$$\frac{\mathrm{d}S_{i,a}}{\mathrm{d}t} = S_{d,i}(t_0) - S_{d,i}(t_1) \tag{A3}$$





3. Calculate annual transpiration following the water balance.

$$\overline{E}_{t,a} = \overline{P}_{e,a} - \overline{Q}_a - \frac{dS_a}{dt} \tag{A4}$$

4. Calculate $c_a$ for each hydrological year based on the annual $E_t$ estimate from step 3 and calculate daily $E_t$.

$$c_{i,a} = \frac{\overline{E}_{t,i,a}}{\overline{E}_{p,a}} \tag{A5}$$

$$E_{t,i}(t) = c_{i,a} E_p(t) \tag{A6}$$

5. Calculate storage deficits based on daily $E_t$ from step 4.

$$S_{d,i} = \max(0, -\int_{1973}^{2010} (P_e - E_{t,i}) dt) \tag{A7}$$

6. The input storage deficit of iteration $i+1$ in step 2 is the average of iteration $i$ and $i-1$

$$S_{d,i+1} = \frac{S_{d,i} + S_{d,i-1}}{2} \tag{A8}$$

The following three constraints are set to the iterations:

– The long term water balance closes ($\overline{P}_e - \overline{Q} - \overline{E}_t \approx 0$).

– Annual transpiration is always larger than zero and smaller than the annual potential evaporation.

– Variations in $c$ are limited by $c_{0,a} - f\,c_{0,a} < c_{i,a} < c_{0,a} + f\,c_{0,a}$ with $f$ a coefficient set to 0.25.

Figure A1 illustrates the iterative approach for storage deficit calculations. Daily $P$, $E_p$ and $E_t$ based on Eq. (A1) are presented in Fig. A1a. Figure A1b shows annual variations of $P_e$ and $E_t$. During the years 1980-1984 $P_e$ is clearly less than average and $E_{t,0}$ estimate is likely too high in these years because vegetation has less water available for transpiration this year. The final iteration $E_{t,9}$ provides a more realistic inter-annual pattern of transpiration. Initial and final iteration storage deficits are presented in Fig. A1c.

*Author contributions.* The study was conceived by RE and amended with input from all authors. FO carried out the study, analysed the results and wrote the manuscript with input and feedback from RE, MH and AA. Specific knowledge and support for the $S_r$ calculations were provided by MH and specific knowledge and support for the EC-EARTH model were provided by AA.

*Competing interests.* The authors declare that they have no conflict of interest





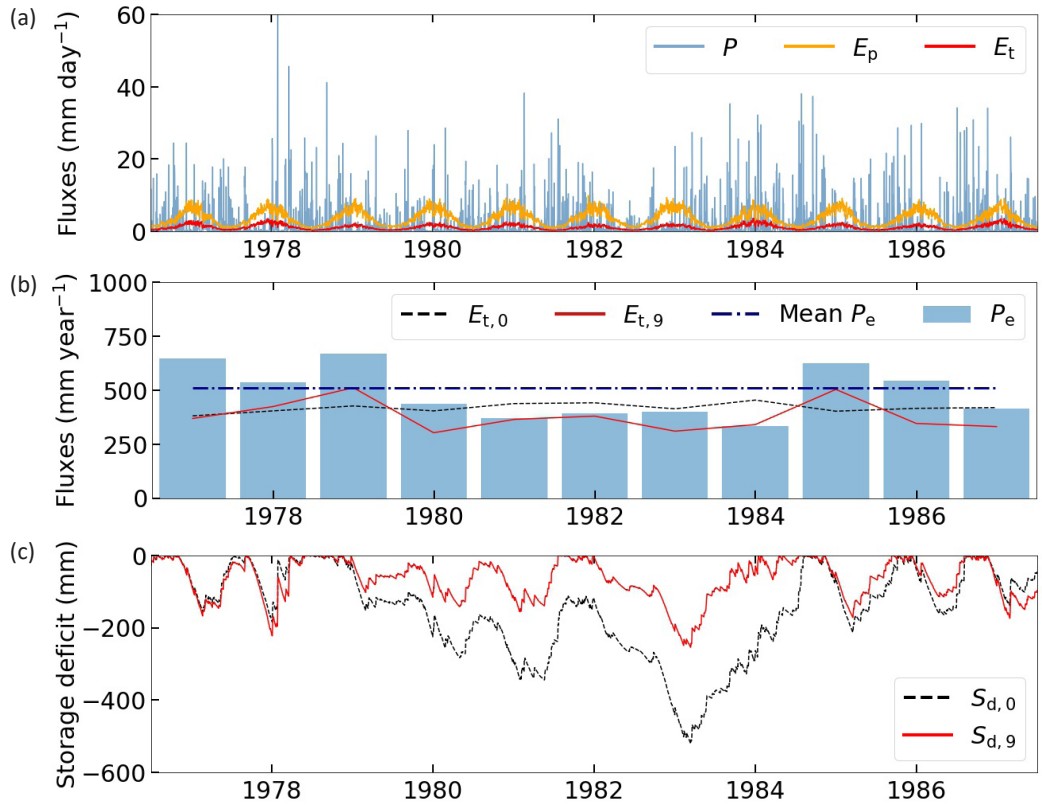

**Figure A1.** Storage deficit iteration approach in a temperate catchment for the time period 1977–1987. (a) Daily water fluxes with $P$ precipitation, $E_\mathrm{p}$ potential evaporation and $E_\mathrm{t}$ the initial transpiration calculation based on Eq. (6); (b) Annual water fluxes with $P_\mathrm{e}$ effective precipitation, $E_\mathrm{t,0}$ the initial transpiration estimate and $E_\mathrm{t,9}$ the final iteration transpiration estimate. Mean $P_\mathrm{e}$ is based on the full time period (1973-2010); (c) Daily storage deficit with $S_\mathrm{d,0}$ the initial calculation and $S_\mathrm{d,9}$ the final iteration.

*Acknowledgements.* Acknowledgement is made for the use of ECMWF's computing and archive facilities in this research that were provided by the KNMI and by ECMWF in the framework of special project SPITALES. This work was supported by the European Union's Horizon 2020 research and innovation programme under grant agreement N. 101004156 (CONFESS project). RE acknowledges funding from the Netherlands Organization for Scientific Research (NWO), project number 016.Veni.181.015.






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
