# Peer review of "Climate controlled root zone parameters show potential to improve water flux simulations by land surface models"

_Earth System Dynamics, 2021_

## Referee Comment (RC2)

**Manuscript:** Climate controlled root zone parameters show potential to improve water flux simulations by land surface models

**Major remarks**

The authors analysed the effect of using a climate dependent root zone storage capacity $S_r$ instead of a vegetation type dependent $S_r$ on simulated runoff and evaporation fluxes in Australia. They estimated this 'climate controlled' $S_r$ with the "memory method" (MM) in which $S_r$ is derived from the vegetation's memory of past root zone water storage deficits and introduced this into the HTESSEL land surface scheme. By using forcing from the GSWP-3 dataset, the new $S_r$ led to improved seasonal climatologies (1975–2010) and inter-annual anomalies of river discharge over 15 selected small catchments while only a negligible impact on evaporation fluxes and long-term mean model biases was found. As the climate control on root development is not regarded by most of the existing land surface models (LSMs), this study is a valuable contribution on climate – hydrology interactions within the topic of Earth System Modelling.

My only major remark is that I miss a more thorough analysis on why the HTESSEL evaporation is rather insensitive to the changes in $S_r$. Opposite to the present study, the evaporation of other LSMs reacts usually more sensitive to water holding capacity changes. However, many climate models tend/tended to have LSMs with more shallow soils, and, hence lower $S_r$, so that in related studies, $S_r$ was often increased. In the present study, CTR seem to have a rather large $S_r$, and there is a general reduction of $S_r$ using the MM method. Does this has something to do with this insensitivity?

In addition, the authors state that E is rather insensitive to changes in SR because E depends on the relative soil moisture. It is well known that E is sensitive to soil moisture when soil moisture in the transitional regime between the wilting point soil moisture (dry regime if moisture is below) and a critical soil moisture above which evapotranspiration is occurring at its potential rate $E_{pot}$ (wet regime). In order to investigate this further I suggest considering in which catchments, the soil moisture is in the transitional regime, and whether the relative soil moisture changes due to the introduction of the new $S_r$. If, for example, a catchment is in the wet or dry regime for most of months, then E will not react to changes in $S_r$.

The paper is generally written well so that I suggest accepting the paper for publication after minor revisions have been conducted.

**Minor remark**

In the following suggestions for editorial corrections are marked in *Italic*.

p. 1 - line 17
… long-term annual mean *river discharge* are …

p. 7 - line 146
It is written:
… long-term mean transpiration derived from the water balance ($E_t = P_e - Q$) …

The evapotranspiration at the land surface (without the canopy, such as in your balance equation) comprises also evaporation of snow and evaporation over bare soil. While the first may not play a role over Australia, the latter certainly does as I do not expect that all catchments are completely covered by vegetation so that bare soil fraction equals Zero. Please elaborate on this issue in more detail.

p. 10 - line 198
It is written:
Total discharge (Q) is the sum of $Q_s$ and $Q_{sb}$ (Eq. 13).

Do you consider lateral flow within the catchment and the respective delay due to lateral transport? Or are the catchments small enough so that this delay is negligible. Please comment!

p. 10 - line 217
… would *cause* the model …

p. 11 - line 221
It is written:
It should be noted that the layer depths for thermal diffusion calculations are not modified in the MD model.

Do you assume bedrock (i.e. zero moisture) below the root zone for the thermal calculation if z4 is reduced for water? What do you do? How does this affect your simulation?

p. 13 - line 281
… affected as *shown* in …

p. 17 - line 324-325
… related to *the applied methodology* which will be further discussed in Section *4.2.*

As Section 4.2 is about 'Methodological uncertainty', I assume you point to Sect. 4.2, and not 4.3 as written in the manuscript?!

p. 19 - line 379-380
…is *only a* function …

p. 19 - line 385
It is written:
$S_{r,CTR}$ was found to be considerably larger than the climate controlled estimate $S_r$ …

How do these values compare to those of other LSMs? This comment is also related to my major remark.

p. 19 - line 403-404
It is written:
On the other hand, surface and subsurface runoff depend on the cumulative moisture content of the soil at any given time.

What do you mean with "cumulative" content? Looking at Figure 4, I assume that both depend on the moisture content above a certain threshold. Please clarify!

---

## Author Comment (AC1)

**Referee 1**

*'This is definitely an important study. I agree that Land Surface Model could be improved in the definition of soil depth. The impact of rooting depth in determining the active soil moisture and the amount of water that can transpire ultimately influence the occurrence and amplification of heat waves, an increasingly pressing issue in an era of climatic changes. The model development proposed in this study could improve climate services that are primary forms of climate adaptation in many sectors.*

*The manuscript is generally well-written and contains high-quality research. However, I was not familiar with the method used in this manuscript. This is reflected in some questions and comments that the author could consider.'*

We would like to thank the referee for the comments. We appreciate the time and effort taken to read our manuscript in detail and to provide us the very useful and interesting thoughts on our research. We will take the comments into account in revising the manuscript.

We have separated the different comments (shown in *italic*) and have written our replies below. Text in the original manuscript is shown in *'italic'* and revised text in '**bold**'. Wherever line numbers are mentioned in our reply they refer to the original manuscript version.

*Comment 1.1*

*'The Memory Method is definitely very interesting. I believe it is reasonable enough to assume that local vegetation adapts the rooting depth according to the drought frequency. However, it's difficult to me to understand the way it is implemented in the model, i.e. varying the total soil depth in the model grid cells. I think it would have been more reasonable to change the rooting distribution Z from the model formulation instead of varying the bottom soil thickness. After all, in reality it is the vegetation adapting the roots, not the soil changing thickness. Why this approach was not followed?'*

It is indeed true that in reality vegetation adapts the roots according to drought frequency, and not the soil changing its thickness. The referee suggests an approach to modify the rooting distribution rather than the soil depth.

It should be noted that the soil depth in HTESSEL does not represent the actual soil thickness but instead it represents the hydrologically active depth of the soil as resulting from the actual depth that is reached by the vegetation roots for transpiration. Consistently, the water content in the soil corresponds with the field capacity (i.e. water accessible by vegetation for transpiration), which – excluding wilting point – coincides with root zone storage capacity ($S_\mathrm{r}$). We will add a sentence in the model description of HTESSEL (section 2.4) to stress this.

Transpiration is mediated by the amount of water in the soil that can be effectively accessed by vegetation. In the HTESSEL model this depends on three factors: 1) the total amount of water available in the hydrologically active soil (controlled by total soil depth) 2) the relative depth of the individual soil layers and 3) the rooting distribution (i.e. relative density) in each soil layer.

We acknowledge that we could as well have changed the rooting distribution and the individual soil layer depths. However, in this study we did not want to change multiple model parameters at the same time

to avoid difficulties in identifying the differences between CTR and MD model output. This study is focused on the effect of the total amount of water in the soil available for vegetation transpiration (as controlled by total soil depth). We will add a sentence explaining this choice better in section 2.5.

In follow-up studies we will consider the effects of changing the rooting distribution and the relative depth of soil layers and we will elaborate on this in the discussion (section 4.3).

We will modify L186 in section 2.4 as follows:

 *'...with z **the hydrologically active depth, that corresponds to** the combined depth of all soil layers with roots...'*

We will modify L222 in section 2.5 as follows:

 *'... in Fig. 4. **Also, the root distribution is not modified in MD, because we aimed for a physical representation of Sr (Eq. 22) and we did not want to change multiple model parameters at the same time.**'*

*Comment 1.2*

*'The modelled approach assumes that the maximum holding capacity should be equal to Sr. If I understand well it should rather a minimum value corresponding to dry years right? For tall vegetation, the root depth is defined through the memory method as a function of the 40 years return period drought and 2 years for low vegetation. With the implementation considered in this study it seems like the soil cannot hold more moisture than the one available in dry years. Maybe this is the reason of the apparent systematic underestimation of Sr by the model.'*

If we understand correctly, the referee's perception was that $S_r$ represents the maximum available moisture during dry years, and is therefore a lower limit of soil moisture holding capacity. On the contrary, $S_r$ is defined by the soil moisture deficit, that maximizes during dry years, and, therefore, represents an upper limit of root zone storage. The apparent underestimation of $S_{r,MM}$ compared to $S_{r,CTR}$ is therefore not related to the available moisture in dry years, but related to the moisture deficit in dry years.

We will clarify the relation of soil moisture storage and $S_r$ in the methods chapter in lines 127-130 as follows:

 *' ...is estimated based on catchment hydrometeorological data, according to the methodology described in the studies of De Boer-Euser et al. (2016), Nijzink et al. (2016) and Wang-Erlandsson et al. (2016). **$S_{r,MM}$ is based on an extreme value analysis of the water storage deficit in the vegetation's root zone ($S_d$). $S_d$ maximizes during dry periods, and, therefore, $S_r$ represents an upper limit of root zone storage assuming that vegetation has sufficient access to water to overcome these dry periods.** The cumulative...'*

*Comment 1.3*

*'Also, the definition of Sr for the observations is a linear superposition of different values that are obtained for high and low vegetation, suggesting the these values are different. Modelled Sr is defined as one value for all vegetation types. So not only the implementation is the model seem to assume that the soil cannot hold more water that than in dry years, but also that is does not depend on the vegetation type. Again, maybea different approach is needed.'*

HTESSEL does not implement any sub-grid heterogeneity in the soil discretization and so it does not allow to change soil depths differently for different vegetation types (Fig. 4b)*.* Accordingly*,* $S_{r,MM}$ was defined as one value for all vegetation types (see Eq. 7). The memory method allows us to make a separation in $S_r$ for high and low vegetation, but we combined $S_r$ values for high and low vegetation to get unique catchment-representative $S_r$ and so the MD soil depth consistently with the HTESSEL model formulation. However, we acknowledge that we could get a separation between high and low vegetation $S_r$ by modification of the root distribution separately for high and low vegetation. However, as mentioned before, we did not want to change both root distribution and model soil depths at the same time in this study (see also reply to comment 1.1).

We will clarify this in section 2.3 line 155 as follows:

*'...maximum storage deficits.* **Theoretically we could treat $S_r$ separately for high and low vegetation in HTESSEL, however, this would require changing the root distributions (see section 2.4), which we decided not to do as we did not want to change multiple parameters at the same time.***'*

*Comment 1.4*

*'Finally, I think the modelled Sr could be computed exactly in the same manner as it is done for observations. I understand the author prefer to use a more physical definition, but probably computing it in the same way as in the observation would be a fairer comparison that could be used to better calibrate the modified model.'*

Indeed, we did choose a physically based approach (changing the model soil depths) for implementing $S_r$ as we aimed to investigate if we could clearly observe effects of the memory method $S_r$ estimates on the modelled fluxes in HTESSEL.

However, it would indeed be useful to make a comparison based on soil moisture deficits and the model's effective $S_r$ ($S_{r-eff}$). Thus, following the suggestion of the referee, we explored calculating $S_{r-eff}$ based on the modelled soil moisture deficits and a similar extreme value calculation as was done in the memory method. We will add a line to the HTESSEL model description (2.4), we will change lines 351-352 in the manuscript and we will include Figure C1 in the supplementary material, together with a short discussion of the results shown in this figure.

We will modify line 189 (section 2.4) as follows:

*'is not accessible to roots ...* **It should be noted that we aimed for a physical definition of $S_{r,CTR}$, but that the effective water used by vegetation may be different. We come back to this point more elaborately in the discussion.***'*

We will modify lines 351-352 in section 4.3 as follows:

L351-352: *'This formulation represents the theoretical $S_{r,CTR}$, but it might not fully correspond to the soil moisture in the four layers that is actually used by the modeled vegetation. **The effective $S_r$ is derived from modelled soil moisture storage deficits ($S_{r,CTR-eff}$) and is smaller than $S_{r,CTR}$ based on depths (Fig C1c). This is likely due to the relatively small root percentage in layer 4 prescribed from look-up tables in this layer for most vegetation types compared to the other layers. However, the $S_{r,MM}$ we implemented in the MD model by changing soil depths is close the $S_{r,MD-eff}$ based on modelled soil moisture deficits in the MD model (Fig. C3d).***

[Figure]

*Figure C1 (new Figure S3). Model $S_r$ analysis. (a) $S_{r,MM}$ from the memory method vs. $S_{r,CTR}$ based on HTESSEL soil depth. (b) $S_{r,MM}$ from the memory method vs. $S_{r,CTR-eff}$ based on modelled soil moisture deficits. (c) $S_{r,CTR}$ based on soil depth vs. $S_{r,CTR-eff}$ based on modelled soil moisture deficits. (d) $S_{r,MM}$ from the memory method vs. $S_{r,MD-eff}$ based on modelled soil moisture deficits.*

*Minor comments*

*Comment 1.5*

*'Equation 2 is formally incorrect. The time-dependency S(t) does not match the right-handside where t is the integrating variable that should disappear after the integration is performed. The actual variable that survives should be related to t0 and t1, which are notdefined either than in Appendix A. I think a subscript for the hydrological year should be preferred in that case. About the subscripts, here 'd' is used. 'r' in used elsewhere, why? I think 'd' stands for deficit and 'r' stand for roots. If so, is the right hand side of equation 7 there should be 'd', not 'r'.'*

We will change Eq. 2 from:

$$S_{\mathrm{d}}(t) = \max(0, -\int_{t_0}^{t_1}(P_{\mathrm{e}} - E_{\mathrm{t}})\mathrm{d}t)$$

to:

$$\boldsymbol{S_{\mathrm{d}}(t) = \max(0, -\int_{t_0}^{\tau}(P_{\mathrm{e}} - E_{\mathrm{t}})\mathrm{d}t)}$$

**with an integration from $t_0$ that corresponds to the first day in the hydrological year 1973 to $\tau$ that corresponds to the daily time steps ending at the last day of the hydrological year 2010.**

There is also confusion about the use of the subscripts 'd' and 'r', in which 'd' stands for deficit, and 'r' for root zone. There is an important difference between $S_{\mathrm{d}}$ and $S_{\mathrm{r}}$: $S_{\mathrm{d}}$ is the storage deficit over time, defined as the cumulative difference between $P$ and $E_{\mathrm{t}}$. On the other hand, $S_{\mathrm{r}}$ is the root zone storage capacity that is calculated applying a Gumbel extreme value analysis to annual maximum storage deficits (see lines 154-155). We will clarify this in lines 127-130 as follows:

*'$S_{r,MM}$ is estimated based on catchment hydrometeorological data, according to the methodology described in the studies of De Boer-Euser et al. (2016), Nijzink et al. (2016) and Wang-Erlandsson et al. (2016). $\boldsymbol{S_{r,MM}}$ is based on an extreme value analysis of the annual maximum water storage deficits in the vegetation's root zone ($\boldsymbol{S_d}$).'*

*Comment 1.6*

*'I agree that considering constant ratio of actual vs. potential evapotranspiration is a crude approximation, especially in water limited regions. I also agree that the other factors mentioned by the authors (groundwater, irrigation) are important as well. These are difficult to improve in the model in short times. However, the author of implemented an iterative step in their approach to reduce strongly the uncertainty relate to the inter-annual variability of that ratio (Appendix A). Couldn't they use somehow the observed evaporation to further improve the estimation? I think this would improve the estimated Sr, especially regarding the intra-seasonal variability, and eventually improve the modified model calibration.'*

The estimation of transpiration in the memory method is solely based on discharge, precipitation and potential evaporation data. This was done because we consider these data reliable for the catchment scale. On the other hand, actual evaporation data (FLUXCOM in this study) is less reliable for the catchment scale, because it is based on point scale estimates of FLUXNET stations that are located far

from the study catchments (Fig. 3 and lines 114-117). Therefore, we decided not to use the FLUXCOM intra-seasonal variability for the $S_{r,MM}$ estimates.

Furthermore, the general finding that $S_{r,MM}$ is considerably smaller than $S_{r,CTR}$ does not change when we use FLUXCOM evaporation in the memory method (average $S_r$ of 284 mm with FLUXCOM, average $S_r$ of 333 mm with scaling $E_p$). As the differences are small, no large effects on the MD model performance are expected.

*Comment 1.7*

*'To the uncertainties, I would add the model drainage rate that, in the current framework, it could as important as the rooting depth. The results obtained by the authors could be due to excessive retention (slow drainage) rather than too deep root zone. This is also supported by the fact that other models are rather augmenting the soil depth adding a groundwater layer instead of reducing it (e.g. CLM).'*

We agree that a groundwater layer is important for modelling the base flow and that the lack of a groundwater layer in HTESSEL causes uncertainty in the modelled river discharge. However, in this paper we aimed to improve the representation of the vegetation's root zone, and an analysis of modelled base flow and the potential of an additional groundwater reservoir was out of scope.

We will add this uncertainty to the discussion in 4.3.

*Comment 1.8*

*'Equation 22: where is the equal sign?'*

There is confusion about Eq. 22:

$$S_{r,MD} = S_{r,MM} = z_{MD}(\theta_{cap} - \theta_{pwp})$$

We will change the equation and line 216 to:

$$\mathbf{'}S_{\mathbf{r,MM}} = z_{\mathbf{MD}}(\boldsymbol{\theta}_{\mathbf{cap}} - \boldsymbol{\theta}_{\mathbf{pwp}})$$

with $z_{\mathrm{MD}}$ **the total soil depth in the MD model modified to satisfy** $S_{r,MD} = S_{r,MM}.\mathbf{'}$

Comment 1.9

*'Figure 6: Shouldn't Q + E equal P in the long period? If this should be true, the average anomalies of the modified model on the top panel should be equal the average anomalies of the bottom panels as precipitation does not change, but this is not true. E anomalies are much smaller, Why? Am I missing something?'*

In the long term the water balance closes and Q + E equals P. In Figure 6 the mean values of Q + E for all the three presented bars (observed, CTR and MD) sum up to P. However, the y-axis scale for Q in Figures 6b and 6c is different than in the other subplots, which apparently confuses the reader. Dr. Guswa (referee 3) suggested to use the same y-axis scale in Figures 6b and 6c, and add zoomed figures of the temperate and Mediterranean results to make the plots readable (comment 3.5). Figure C2 is the new version of Figure 6.

[Figure]

*Figure C1(new Figure 6). Monthly seasonal climatology of observed discharge (Q) (top) and FLUXCOM-WB evaporation ($E_{FLUXCOM-WB}$) (bottom) and modeled values in the HTESSEL CTR and MD versions, averaged for the tropical (a, d), temperate (b, e) and Mediterranean (c, f) catchments for the time series 1975-2010. b1 and c1 represent the same data as b2 and c2, but with a different y-axis. Similar figures for the individual catchments are shown in Fig. S1 (Q) and Fig. S2 (E).*

---

## Author Comment (AC2)

**Referee 2 (Stefan Hagemann)**

*'The authors analysed the effect of using a climate dependent root zone storage capacity Sr instead of a vegetation type dependent Sr on simulated runoff and evaporation fluxes in Australia. They estimated this 'climate controlled' Sr with the "memory method" (MM) in which Sr is derived from the vegetation's memory of past root zone water storage deficits and introduced this into the HTESSEL land surface scheme. By using forcing from the GSWP-3 dataset, the new Sr led to improved seasonal climatologies (1975–2010) and inter-annual anomalies of river discharge over 15 selected small catchments while only a negligible impact on evaporation fluxes and long-term mean model biases was found. As the climate control on root development is not regarded by most of the existing land surface models (LSMs), this study is a valuable contribution on climate – hydrology interactions within the topic of Earth System Modelling. The paper is generally written well so that I suggest accepting the paper for publication after minor revisions have been conducted.'*

We would like thank dr. Hagemann for the comments. We appreciate the time and effort taken to read our manuscript in detail and to provide us the very useful and interesting thoughts on our research. We will take the comments into account in revising the manuscript.

We have separated the different comments (shown in *italic*) and have written our replies below. Text in the original manuscript is shown in *'italic'* and revised text in '**bold**'. Wherever line numbers are mentioned in our reply they refer to the original manuscript version.

*Comment 2.1*

*'My only major remark is that I miss a more thorough analysis on why the HTESSEL evaporation is rather insensitive to the changes in $S_r$. Opposite to the present study, the evaporation of other LSMs reacts usually more sensitive to water holding capacity changes. However, many climate models tend/tended to have LSMs with more shallow soils, and, hence lower $S_r$, so that in related studies, $S_r$ was often increased. In the present study, CTR seem to have a rather large $S_r$, and there is a general reduction of $S_r$ using the MM method. Does this has something to do with this insensitivity?'*

Indeed, the limited sensitivity of evaporation to changes in soil depth can be partly explained by $S_{\mathrm{r,CTR}}$ being larger than $S_{\mathrm{r,MD}}$. We further elaborate on this in comment 2.2.

*Comment 2.2*

*'In addition, the authors state that E is rather insensitive to changes in Sr because E depends on the relative soil moisture. It is well known that E is sensitive to soil moisture when soil moisture in the transitional regime between the wilting point soil moisture (dry regime if moisture is below) and a critical soil moisture above which evapotranspiration is occurring at its potential rate Epot (wet regime). In order to investigate this further I suggest considering in which catchments, the soil moisture is in the transitional regime, and whether the relative soil moisture changes due to the introduction of the new Sr. If, for example, a catchment is in the wet or dry regime for most of months, then E will not react to changes in Sr.'*

Figure C3 shows modelled transpiration and soil moisture in the four separate soil layers in a tropical (a), temperate (b) and Mediterranean (c) catchment. In order to further explain the evaporation (in)sensitivity we will consider a wet period (mid 1990) and a dry period (start 1991) in the temperate catchment (Fig. C3b).

During the wet periods soil moisture in the upper three layers is above or close to $\theta_{cap}$ and little differences are observed between CTR and MD. However, in layer 4, MD soil moisture is larger than CTR soil moisture. In this case evaporation is not moisture limited and is controlled by the top three layers because of the larger root distribution in these layers (eq. 14 and 15). Therefore, the modelled transpiration is not sensitive to the increase in layer 4 soil moisture in MD.

During the transition from a wet to a dry period, the upper three layers dry out first, as there is a reduction in precipitation input. As these layers are dry, evaporation is controlled by the fourth layer. Layer 4 soil moisture in MD also reduces to values close to $\theta_{pwp}$, while in CTR layer 4 remains wet. This difference causes the sensitivity of transpiration in MD during this wet to dry transition.

Most of the time the modelled soil moisture is in the wet and insensitive regime, and, therefore, the overall effects of MD on modelled evaporation are small.

We will add this explanation and Figure C3 to the revised manuscript in the discussion 4.1.

[Figure]

*Figure C3 (new Figure 8). Modelled transpiration and soil moisture with CTR and MD models in a (a) tropical (Mi), (b) temperate (Na) and (c) Mediterranean (K) catchment. From top to bottom: transpiration, relative difference between CTR and MD transpiration* $\left(\frac{E_{t,CTR}- E_{t,MD}}{E_{t,CTR}}\right)$, *soil moisture layer 1, soil moisture layer 2, soil moisture layer 3, soil moisture layer 4. Additionally, the vegetation coverage ($C_L$ and $C_H$) and the relative rooting distribution ($R_k$) for the dominant high and low vegetation types are presented.*

*Minor remark*

*Comment 2.3*

*'p. 1 - line 17*

*… long-term annual mean river discharge are …'*

Ok, we will change this line.

*Comment 2.4*

*'p. 7 - line 146 It is written:*

*… long-term mean transpiration derived from the water balance (Et = Pe −Q) …*

*The evapotranspiration at the land surface (without the canopy, such as in your balance equation) comprises also evaporation of snow and evaporation over bare soil. While the first may not play a role over Australia, the latter certainly does as I do not expect that all catchments are completely covered by vegetation so that bare soil fraction equals Zero. Please elaborate on this issue in more detail.'*

The transpiration flux ($E_\mathrm{t}$) considered here includes both transpiration and soil evaporation. We can defend this linguistic inexactness since $E_\mathrm{soil}$ << $E_\mathrm{t}$, especially during dry periods where the soil deficits are largest and determinant thus of $S_\mathrm{r}$. To illustrate this: soil evaporation only occurs over the top few cm of the soil (in HTESSEL only top layer of 7 cm and ranging from top 4 cm – top 15 cm soil depending on soil characteristics as found in Wythers et al. (1999)). Considering top 7 cm of soil active for soil evaporation, this translates into an $S_\mathrm{r}$ of 70 mm * 0.2 (approximation of plant available water) → 14 mm. This is an order of magnitude smaller than the $S_\mathrm{r}$ estimates from the memory method which are on average 333 mm.

We will change line 146 as follows:

*'…long-term mean transpiration derived from the water balance ($E_t = P_e − Q$) and $E_p$ (mm year$^{-1}$) the long-term mean potential evaporation. $E_t$ **considered here includes both transpiration and soil evaporation, but as $E_{soil}$ << $E_t$, we use the term transpiration for simplicity.'***

Reference: *Wythers, K.R., Lauenroth, W.K. and Paruelo, J.M. (1999), Bare-Soil Evaporation Under Semiarid Field Conditions. Soil Sci. Soc. Am. J., 63: 1341-1349. https://doi.org/10.2136/sssaj1999.6351341x*

*Comment 2.5*

*'p. 10 - line 198 It is written:*

*Total discharge (Q) is the sum of $Q_S$ and $Q_{Sb}$ (Eq. 13).*

*Do you consider lateral flow within the catchment and the respective delay due to lateral transport? Or are the catchments small enough so that this delay is negligible. Please comment!'*

We do not consider lateral flow within the catchment and the respective delay due to lateral transport. Our analysis is based on monthly fluxes, and routing at this timescale becomes negligible for the size of our catchments. To illustrate this: for a catchment size of 2500 km² (50 km x 50 km) and a 1 m/s (= 86.4 km/day) flow, the water will flow through the entire catchment within approximately 1 day, so we can neglect routing at monthly time scales.

We will change line 198 as follows:

*'Total discharge (Q) is the sum of Qs and Qsb (Eq. 13)* **and typical travel times through the catchments are about 1 day at most, we did not consider routing to be important at the monthly time scale for which we analyze the results.'**

Ok, we will change this line.

*Comment 2.7*

*'p. 11 - line 221 It is written:*

*It should be noted that the layer depths for thermal diffusion calculations are not modified in the MD model.*

*Do you assume bedrock (i.e. zero moisture) below the root zone for the thermal calculation if z4 is reduced for water? What do you do? How does this affect your simulation?'*

HTESSEL defines depth parameters for moisture calculations and for thermal calculations separately. The layer depths for thermal diffusion calculations are kept constant in CTR and MD, but layer depths for moisture calculations are modified in MD. We assume zero moisture below the root zone for thermal calculations if z4 is reduced for water. The thermal calculations only interact with soil moisture calculations by the relative soil moisture content in a layer. Therefore, the depth change does not directly affect thermal calculations. We found insensitivity of the soil layer temperatures to depth changes in MD.

We will change line 221 as follows:

'... are not modified in the MD model **and that the soil layer temperatures are insensitive to depth changes in MD.**'

*Comment 2.8*

*'p. 13 - line 281*

*… affected as shown in …'*

Ok, we will change this line.

*Comment 2.9*

*'p. 17 - line 324-325*

*… related to the applied methodology which will be further discussed in Section 4.2.*

*As Section 4.2 is about 'Methodological uncertainty', I assume you point to Sect. 4.2, and not 4.3 as written in the manuscript?!'*

Yes, we will change this.

*Comment 2.10*

*'p. 19 - line 379-380*

*…is only a function …'*

Ok, we will change this line.

*Comment 2.11*

*'p. 19 - line 385 It is written:*

*$S_{r,CTR}$ was found to be considerably larger than the climate controlled estimate $S_r$ …*

*How do these values compare to those of other LSMs? This comment is also related to my major remark.'*

It is expected that our $S_{r,MM}$ estimates relate differently to $S_r$ values in other LSMs. This paper focuses on the case of HTESSEL and analyzing other LSMs was out of scope. We will change line 385 as follows:

*'… $S_{r,CTR}$ was found to be considerably larger than the climate controlled estimate $S_{r,MM}$ in 14 out of 15 catchments. **These findings could be different for other LSMs when they have more shallow soil depths**.'*

*Comment 2.12*

*'p. 19 - line 403-404 It is written:*

*On the other hand, surface and subsurface runoff depend on the cumulative moisture content of the soil at any given time.*

*What do you mean with "cumulative" content? Looking at Figure 4, I assume that both depend on the moisture content above a certain threshold. Please clarify!'*

With cumulative content, we mean the total, absolute moisture content of the soil, which is equal to the relative soil moisture multiplied with the soil depth. Changing the depth of the modelled soil therefore more strongly affects the quantity of surface and subsurface runoff fluxes.

We will change line 403-404 as follows:

*'On the other hand, surface and subsurface runoff depend on the **total** moisture content of the soil at any given time.'*

---

## Author Comment (AC3)

**Referee 3 (Andrew Guswa)**

*'In this work, the authors compare results from the HTESSEL land-surface model with vegetation root depth determined in two different ways. The control case (CTR) comprises root depths (and, correspondingly, root-zone storage capacity, Sr) determined via soil depth; in another formulation (MD), Sr is determined via the memory method, which is based on the concept that plant roots adjust to mitigate droughts with certain return periods. These models are implemented for 15 catchments in Australia with tropical, temperate, and Mediterranean climates.*

*Results reveal that the root-zone storage capacities determined via the memory method are shallower and more variable across the watersheds than those from the control cases. This is in contrast to what others have suggested – that root depth in LSMs may be too small. The changes in Sr (from the CRT to MD approach) manifest as improvements in the variability and seasonality of streamflow. Long-term water balances and ET are relatively insensitive to changes in Sr. The paper is well-written, and the methods and results are well-explained and valuable. I offer a few suggestions and comments below, and I recommend publication with minor revision.'*

We would like thank dr. Guswa for the comments. We appreciate the time and effort taken to read our manuscript in detail and to provide us the very useful and interesting thoughts on our research. We will take the comments into account in revising the manuscript.

We have separated the different comments (shown in *italic*) and have written our replies below. Text in the original manuscript is shown in *'italic'* and revised text in '**bold**'. Wherever line numbers are mentioned in our reply they refer to the original manuscript version.

*Comment 3.1*

*Low sensitivity of ET*

*'The authors note that model estimates of ET do not appear very sensitive to the differences in Sr. I agree with Hageman that the low sensitivity may be due in part to the fact that the values of Sr,MD are less than Sr,CTR. The catchments in question are arid/water-limited, and ET dominates the water balance, with annual ET being 4-16 times the annual streamflow. One of the proclaimed advantages of the memory method is that it facilitates deep and/or expansive roots so as to maintain ET during periods of drought. In this work, it seems that the soil-based root depths (Sr,CTR) were already sufficiently large, such that the limits of the root zone storage were not being approached. An analogy would be a water-supply reservoir that never fills – if you make it a little smaller, you do not affect the available supply, since you were starting with excess capacity. The authors may wish to expand their discussion in the paper along these lines.'*

We investigated this issue (also based on the comments 2.1 and 2.2 by dr. Hagemann). Indeed, the limited sensitivity of evaporation to changes in soil depth can be partly explained by $S_{r,CTR}$ being larger than $S_{r,MD}$. It was found that modelled evaporation is sensitive during dry periods only, when the MD soil moisture in layer 4 is smaller than CTR. However, most of the time soil moisture is in a wet, insensitive regime, and the evaporation does not approach the limits of soil moisture storage. This is illustrated by Figure C3, that shows modelled transpiration and soil moisture in the four separate soil layers in a tropical (a), temperate (b) and Mediterranean (c) catchment. In order to further explain the evaporation

(in)sensitivity we will consider a wet period (mid 1990) and a dry period (start 1991) in the temperate catchment (Fig. C3b).

During the wet periods soil moisture in the upper three layers is above or close to $\theta_{cap}$ and little differences are observed between CTR and MD. However, in layer 4, MD soil moisture is larger than CTR soil moisture. In this case evaporation is not moisture limited and is controlled by the top three layers because of the larger root distribution in these layers (eq. 14 and 15). Therefore, the modelled transpiration is not sensitive to the increase in layer 4 soil moisture in MD.

During the transition from a wet to a dry period, the upper three layers dry out first, as there is a reduction in precipitation input. As these layers are dry, evaporation is controlled by the fourth layer. Layer 4 soil moisture in MD also reduces to values close to $\theta_{pwp}$, while in CTR layer 4 remains wet. This difference causes the sensitivity of transpiration in MD during this wet to dry transition.

Most of the time the modelled soil moisture is in the wet and insensitive regime, and, therefore, the overall effects of MD on modelled evaporation are small.

We will add this explanation and Figure C3 to the revised manuscript in the discussion 4.1.

[Figure]

Figure C3 (new Figure 8). Modelled transpiration and soil moisture with CTR and MD models in a (a) tropical (Mi), (b) temperate (Na) and (c) Mediterranean (K) catchment. From top to bottom: transpiration, relative difference between CTR and MD transpiration $\left(\frac{E_{t,CTR}-E_{t,MD}}{E_{t,CTR}}\right)$, soil moisture layer 1, soil moisture layer 2, soil moisture layer 3, soil moisture layer 4. Additionally, the vegetation coverage ($C_L$ and $C_H$) and the relative rooting distribution ($R_k$) for the dominant high and low vegetation types are presented.

*'Additionally, the authors may also wish to acknowledge (again) the uncertainty in the ETobservations. As they point out in lines 110-115, the estimates from the water balances are lower than the estimates from the FLUXCOM data by 20%, and it may be worth reminding the reader of this in the discussion.'*

We will add the following sentence on the uncertainty of the $E_t$ observations in lines 321 to remind the reader of this:

*'… E was very limited in all climate regions (Table 3; Table 4; Table S6; Fig. S2). **As stated before, the reliability of the FLUXCOM E is questionable in our study catchments (Fig. 3). Although the model performance with respect to E fluxes is uncertain,** the lack of evaporation sensitivity to $S_r$ was unexpected and requires more in depth evaluation of HTESSEL.'*

*Comment 3.3*

*'Lines 110-121 and Figure 1: The authors may wish to address whether the lower estimates of ET via the water balance method might be affected by or attributable to deep groundwater drainage that does not get recorded by the stream gauges.'*

We assumed that the catchments are large enough that deep groundwater drainage does not play a major role in the catchment water balance. Therefore, this would not explain the lower estimates of E-WB compared to E-FLUXCOM.

We will change line 115 as follows:

*'… an integrated catchment scale estimate as it is derived from observations of Q **assuming that the catchments are large enough to neglect deep groundwater drainage to or from other catchments**.'*

*Comment 3.4*

*'Lines 155-eq 7: Like the other reviewers, I am a bit confused by the approximation of "Catchment Sr,MM" by a weighted average of the Sr values for trees and grasses. As drainage below the root zone is a non-linear process, this will affect the results (e.g., drainage below the grass portion may be non-zero, whereas drainage below an "average" root depth may be zero). I understand that the model is limited in its resolution, and one has to make some concessions. It may be worth an expanded comment, however.'*

The memory method allows us to make a separation in $S_r$ for high and low vegetation. We decided to combine $S_r$ values for high and low vegetation to get one catchment representative $S_r$ and consequently one model soil depth in MD. This approach was chosen because the CTR model parameterization does not allow to change soil depths differently for different vegetation types, as HTESSEL has one soil discretization for the entire grid cell (Fig. 4b). However, we acknowledge that we could get a separation between high and low vegetation $S_r$ by modification of the root distribution separately for high and low vegetation, as mentioned in the reply to referee 1 (comment 1.3). However, in this study we did not want to change multiple model parameters at the same time, to avoid difficulties in identifying the differences

between CTR and MD model output. Modification of both rooting distribution and soil layer depth was therefore not desired.

*L 155: '...maximum storage deficits.* **Theoretically we could treat $S_r$ separately for high and low vegetation in HTESSEL, however, this would require changing the root distributions (see section 2.4), which we decided not to do as we did not want to change multiple parameters at the same time.***'*

*Comment 3.5*

*'Figure 6 is a nice figure with a high information density. Like reviewer 1, however, I found the differences in y-axis scales to make interpretation challenging, e.g., the differences in the Temperate and Mediterranean Q appear (visually) to be much greater than those in E. Perhaps it would be worth showing the six subplots, first with all the same y-scale, and to then provide second versions of 6b and 6c that are more zoomed in. I think seeing how very small the runoff is from the Temperate site, before diving into the differences among the models and observations, would help the reader.'*

For visibility of Fig. 6 we will follow your suggestion to use the same y-axis scale in all subplots, and additionally provide 'zoomed' versions of Fig. 6b and 6c. Figure C2 is the new version of Fig. 6. (see also comment 1.9 by referee 1)

[Figure]

*Figure C1 (new Figure 6). Monthly seasonal climatology of observed discharge (Q) (top) and FLUXCOM-WB evaporation ($E_{FLUXCOM-WB}$) (bottom) and modeled values in the HTESSEL CTR and MD versions, averaged for the tropical (a, d), temperate (b, e) and Mediterranean (c, f) catchments for the time series 1975-2010. b1 and c1 represent the same data as b2 and c2, but with a different y-axis. Similar figures for the individual catchments are shown in Fig. S1 (Q) and Fig. S2 (E).*

*Comment 3.6*

*'Lines 298-302: The authors make a point to acknowledge that Sr need not be synonymous with root depth, which is fair enough. In this work, however, it IS synonymous, and I found this introduction to the discussion a bit odd. I recommend that the second sentence of the first paragraph be dropped.'*

It is correct that in HTESSEL the root depth is synonymous with the soil depth, as roots are present over the four model soil layers. However, it should be noted that in $S_r$ represents a conceptual water volume in the model world, without the assumption where this volume is in reality. The model soil depth and root depth are model parameters that are required to schematize the $S_r$ in the model, but are also not found in nature in the way they are schematized in the model.

We will change line 298-302 as follows:

*'… in simulated evaporation (Kleidon and Heimann, 1998; Pan et al., 2020; Sakschewski et al., 2020).* **However, $S_r$ represents a conceptual water volume that is accessible to roots, without the assumption where this volume is in reality.** *Therefore, $S_r$ is not necessarily proportional to root depth as a small $S_r$ does not preclude the presence of deep roots, as illustrated in Fig. 4 in Singh et al. (2020).'*